CellPress

## Perspective

# Bridging genomics' greatest challenge: The diversity gap

Manuel Corpas,[1,2,3,*] Mkpouto Pius,[1] Marie Poburennaya,[4] Heinner Guio,[5] Miriam Dwek,[1] Shivashankar Nagaraj,[6] Catalina Lopez-Correa,[7] Alice Popejoy,[8,9] and Segun Fatumo[10,11]

[1]Life Sciences, University of Westminster, 115 New Cavendish Street, W1W 6UW London, UK
[2]The Alan Turing Institute, London, UK
[3]Cambridge Precision Medicine Ltd., ideaSpace, University of Cambridge Biomedical Innovation Hub, Cambridge, UK
[4]Queen Mary University of London, London, UK
[5]INBIOMEDIC Research and Technological Center, Lima, Peru
[6]Centre for Genomics and Personalised Health, Queensland University of Technology, Brisbane, QLD, Australia
[7]Genome Canada, Ottawa, ON, Canada
[8]Department of Public Health Sciences (Epidemiology), School of Medicine, University of California, Davis, Davis, CA, USA
[9]UC Davis Comprehensive Cancer Center (UCDCCC), UC Davis Health, University of California, Davis, Sacramento, CA, USA
[10]African Computational Genomics (TACG) Research Group, The MRC Uganda Medical Informatics Centre (UMIC), MRC/UVRI and LSHTM, Entebbe, Uganda
[11]Precision Health University Research Institute, Queen Mary University of London, London, UK
*Correspondence: m.corpas@westminster.ac.uk

## SUMMARY

Achieving diverse representation in biomedical data is critical for healthcare equity. Failure to do so perpetuates health disparities and exacerbates biases that may harm patients with underrepresented ancestral backgrounds. We present a quantitative assessment of representation in datasets used across human genomics, including genome-wide association studies (GWASs), pharmacogenomics, clinical trials, and direct-to-consumer (DTC) genetic testing. We suggest that relative proportions of ancestries represented in datasets, compared to the global census population, provide insufficient representation of global ancestral genetic diversity. Some populations have greater proportional representation in data relative to their population size and the genomic diversity present in their ancestral haplotypes. As insights from genomics become increasingly integrated into evidence-based medicine, strategic inclusion and effective mechanisms to ensure representation of global genomic diversity in datasets are imperative.

## BACKGROUND

Providing equitable healthcare that is informed by robust evidence necessitates representation of patient diversity, including genetic ancestry.[1] Using Adsit-Morris et al.'s proposal of equity as "a core principle in governing emerging science and technology,"[2] we evaluate developments in diversity and inclusion of research participants in genomic datasets from different global data resources to characterize representation and infer our current capacity for precision health equity.

While achieving diverse representation in datasets is critical for human health, efforts to diversify participation in genomic research face significant challenges, including a deep-rooted mistrust in the scientific community. This mistrust often stems from past misconduct and unethical practices in research, particularly involving marginalized communities. Historical instances of exploitation, such as the Tuskegee syphilis study[3] and the unauthorized use of Henrietta Lacks's cells,[4,5] have left a legacy of skepticism and wariness toward scientific research among underrepresented populations. Acknowledging and addressing these historical injustices is crucial for rebuilding trust

and fostering greater participation from historically underserved populations.

In 2009, Need and Goldstein[6] published the first quantitative review of ancestral diversity for genome-wide association studies (GWASs). They analyzed raw data downloaded from the GWAS Catalog[7] at the European Bioinformatics Institute-European Molecular Biology Laboratory, which contained free-text descriptions of participant numbers and population labels in GWAS publications. Bustamante et al.[8] popularized Need and Goldstein's finding that 96% of GWASs had been conducted primarily on people of European ancestry. This prompted many GWAS scholars to introduce the now-emblematic sampling bias pie chart (Figure 1, left) to their slide decks, warning audiences to limit applications of their research findings to non-European ancestry groups. These efforts did not, however, lead to widespread changes in GWAS research practice.

Five years later, Popejoy and Fullerton[9] published an update on the lack of ancestral diversity in GWASs, showing that still less than 20% of participants were of non-European ancestry (Figure 1, right), with most growth resulting from an increase in participation in Asian countries. This finding signaled that

progress in our understanding of global genomic diversity and its contributions to health was unacceptably slow. The study also showed that genomics research was being conducted in only a handful of locations worldwide, reflecting the stagnant nature of representation of global populations, which had barely shifted in over a decade. Staff and scholars at the US National Human Genome Research Institute[10] responded to this call to action by describing the benefits and challenges of including diverse participants in genomics research and made recommendations toward achieving greater representation in GWASs.

In a study led by Fatumo and colleagues in 2022,[11] a subsequent analysis of the GWAS Catalog revealed that the vast majority of GWASs were still conducted in people of European descent, with an estimated 72% of participants recruited from just three countries: the United States, the United Kingdom, and Iceland.[12] Through these published investigations, missing diversity in our evidence base for genomic medicine has been recognized as a problem by major biomedical research funders, the pharmaceutical industry, biotechnology companies, and the broader scientific community.[13]

Despite more GWASs being conducted in ancestrally diverse, non-European populations, the total number of GWASs carried out annually has also increased, with many studies using the same European-based datasets, such as the White/British-labeled ($N \sim$425,000) subsample of the UK Biobank.[14] This has led to periodic decreases and stagnation in the proportion of underrepresented populations included in GWASs (e.g., 19% non-European in 2016 to 14% in 2021). The predominance of GWAS publications using the White/British samples from the UK Biobank should therefore be considered when interpreting participant numbers and the representation of genetic diversity in GWASs.

It is also important to note that UK Biobank data contain $\sim$35,000 "non-Europeans,"[15] which are regularly excluded from genetic analyses using this dataset but could be a useful resource for contributing GWAS results from more diverse genetic ancestral backgrounds. The widespread reliance on White/British UK Biobank data for GWASs highlights both the strengths and limitations of using such a centralized and comprehensive dataset. While it enables detailed and extensive genetic research by linking electronic health records to genetic data, it also underscores the need for diverse representation in genomic studies to ensure that findings are applicable to broader populations. Understanding that there may be more to discover in UK Biobank subsamples from other ethnic or ancestral groups than by repeated GWASs for the same traits using White/British samples may yet motivate researchers to conduct analyses with smaller sample sizes, but more predictive power.

## ONGOING EFFORTS TO INCREASE DIVERSITY

Across the globe, significant efforts are being undertaken to enhance diversity in genomics. The All of Us project,[16] for instance, has actively recruited participants from various ancestries across the United States to build one of the most diverse health databases in the world. The Mexico City Prospective Study[17] and the Peruvian Genome Project[18] are defining Latin American initiatives aiming to provide insights into the unique health challenges faced by admixed and native indigenous communities of the Americas. In the Middle East, the Qatar Biobank Cohort Study[19] has broadened the scope of representation for this region. Similarly, the Human Pangenome Project[20] is working on sequencing genomes from historically underrepresented populations. These initiatives have focused on collecting diverse genetic data from underrepresented populations to ensure more inclusive and representative genome database references, which will better inform our current landscape of genomic human variation. The data they are generating are steadily contributing to a more inclusive genomics landscape, although much remains to be done to accelerate the progress and ensure broader global representation.

## ANCESTRY BIASES PERSIST IN GWASs

In recent years we have seen an increased proportion of GWASs reporting "missing" ancestry information.[21,22] This trend should be recognized as a sign of increasing precision and transparency in human genetics research. It reflects a growing understanding that race and ethnicity are distinct from genetic ancestry,[23] which is crucial for accurate data interpretation and representation.[24] We have also seen progress toward more inclusive studies that combine and transcend broad ancestral groupings, moving beyond simplistic racial categorizations to a more nuanced understanding of genetic diversity.[18,25] Furthermore, initiatives such as the African Genome Variation Project[26] and the H3Africa Consortium[27] have significantly expanded the repertoire of available genomic data from modern African populations.[28] These projects aim to understand genetic diversity in different African populations and its implications for human health and disease, thus increasing representation of diverse African ancestries in research.

Despite incremental progress in some projects and areas of human genetics and genomics research, ancestral biases remain and must be accounted for.

Today, updated diversity metrics for published GWAS can be accessed in real time on the GWAS Diversity Monitor[29] without having to conduct laborious analyses from scratch. This interactive tool facilitates exploration and export functions providing images of diversity snapshots of the GWAS Catalog, including maps of research locations and data visualizations for trends in diversity over time (Figure 2).

In 2023, as the proportion of participants of European descent in the GWAS Catalog reached 86.5%, the representation of participants labeled "African" remained unacceptably low, at 0.47% (not including African American- or Afro-Caribbean-labeled samples; Figure 2). These gaps in genetic sampling and the resulting dearth of results derived from most parts of the world suggest that achieving equity is still quite far off.

As of September 2024, the total proportion of participants in the GWAS Diversity Monitor (Figure 3) had <1% representation from any population-labeled groups except Asian (3.96%) and European (94.48%). While we do not suggest that these are appropriate categories by which to group participants in analyses, nor are they genetically coherent or mutually exclusive groupings, they are useful for harmonizing rough, disparate

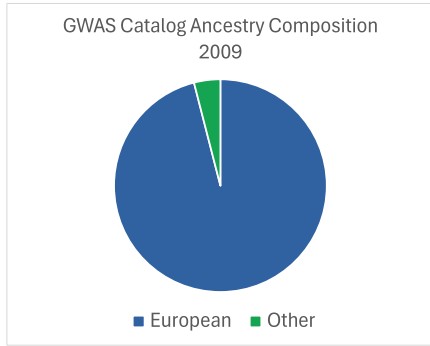
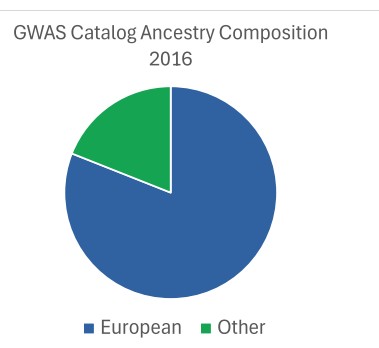

GWAS Catalog Ancestry Composition 2009
■ European ■ Other

GWAS Catalog Ancestry Composition 2016
■ European ■ Other

**Figure 1. Sampling bias**
Left, number of genome-wide association study (GWAS) participants of European ancestry in 2009 from the GWAS Catalog. Right, update by Popejoy and Fullerton (2016)[9] on the ancestry breakdown in GWASs in 2016. Figure adapted from Popejoy and Fullerton.[9]

population descriptors over time to assess equity. To that end, and despite progress being made, the data suggest we continue to fail concerning diversity.

Despite its usefulness, the GWAS Diversity Monitor may report a participant count that can exceed the actual population due to its methodology. Since each individual is counted in every study they are part of, in 2021, the GWAS Diversity Monitor showed 3,675.9 million participants in the United Kingdom, which has a population of 67.0 million. This reflects repeated counts of the same individuals across different traits and phenotypes. Such double counting may make diversity worse than it is, as the absolute number of diverse genomes is increasing.[21]

A major challenge for resolving the insufficiency of genetic diversity included in research is that smaller sample sizes for non-European ancestries and calls for multi-ancestry pooled analyses necessitate combining datasets sampled from different geographic regions to conduct statistical analyses. However, there may be risks associated with omitting ancestry-specific GWASs. Combining datasets from diverse ancestries without properly accounting for differences in sample size may lead to biased results, whereby findings may be skewed toward effects seen in European ancestries due to their relative overrepresentation.[30] This may obscure or prevent the discovery of genetic variations that are not present in European ancestries, despite having strong effects among those who have them.

## IMPLICATIONS OF GLOBAL MISSINGNESS

Underrepresentation of global ancestries is not limited to GWASs. Corpas et al.[31] examined ancestral representation in PharmGKB,[32,33] the leading pharmacogenomics (PGx) database used to document drug-gene interactions. Individuals of European descent represent >63% of all reported population-labeled individuals within PharmGKB. Martin et al.[34] illustrated that polygenic risk prediction algorithms for 17 UK Biobank quantitative traits performed worse for individuals whose ancestries were not well represented in the discovery GWAS that produced the model's input parameters (i.e., effect sizes). These findings suggest that the missingness of global ancestries in GWAS and data that may inform precision medicine will likely impact the development of diagnostic tools and targeted therapies.

Measuring the extent to which missing representation in datasets impacts health and healthcare inequities is inherently chal-

lenging. To quantify relative representation and missingness in global genomic datasets, we need to characterize who is represented more frequently than whom. If we seek to demonstrate bias in who is represented—that is, underrepresentation—then we must use comparative metrics to evaluate whether a particular population grouping (i.e., social categories and/or ancestries described in study populations) is represented more or less often than expected or desired, based on an external threshold. One metric that has been used to conduct such an evaluation is the relative proportion of ancestries represented in the total global population.[34,35]

As a proxy for genetic ancestral backgrounds, biogeographic groupings[36] have been constructed to aid in the categorical assignment of participants reported in PharmGKB. Comparing the proportions of individuals in each biogeographic group to their respective share of the global census population facilitates an estimate of the magnitude of under- and over-representation across these broad geographic categories. Figure 4A shows the relative proportions of study participants represented by each of these biogeographic groupings among all those identified in PharmGKB.[31]

To reflect existing data representation across global populations (Figure 4B), we ascertained populations for each of the biogeographical regions (Table S1), contrasting them with their proportional representation in PharmGKB. We observe a strong European bias in the evidence base generated through PGx research. That is, there is a 46.5% excess of European-ancestry individuals included in this research, based on their overall representation among global populations.

These figures suggest that underrepresentation in PGx is greatest for central/south Asian populations, whose deficit of representation in PharmGKB was estimated at −25.1% from balanced representation, followed by Sub-Saharan African (−14.6%), Latino (−7.8%), Near Eastern (−5.6%), (Indigenous) American (−0.7%), and Oceanian (−0.1%).

## DISPARITIES IN EVIDENCE FOR DRUG EFFICACY AND SAFETY

According to information provided by the US Food and Drug Administration (FDA), an overwhelming 76% of participants in clinical trials between the years 2015–2019 were of primarily European descent.[37] The remaining proportions of ancestries were split, with Asians representing 11% of individuals and Africans or African Americans representing 7% of trial participants (Figure 5). As a result, most data used to inform drug development are likely to be derived from European populations and extrapolated to

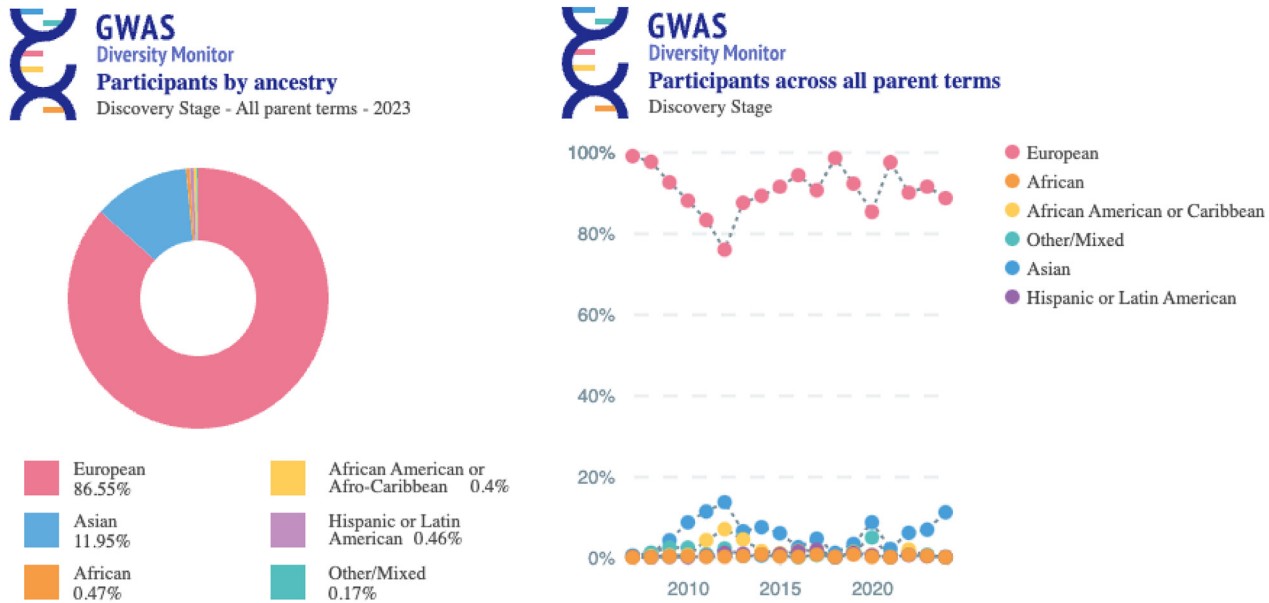

**Figure 2. GWAS Diversity Monitor**
Left, number of GWAS participants by ancestry, including different types of GWASs or health conditions (parent terms), discovery stages, 2023. Right, number of GWAS participants across all parent terms, discovery stage, 2024. Accessed online 9 Sept 2024. We note inconsistencies in labeling and coloring of populations between the figures due to different ways of reporting ancestries by sources.

individuals of other ancestries. It is important to note that these data did not distinguish between African American and Sub-Saharan African, a fact that limits their usefulness in interpreting the magnitude of underrepresentation among biogeographical regions in clinical trials. Only in trials focused on sickle cell disease, tuberculosis, schizophrenia, and onchocerciasis[38] did African and African American individuals exhibit greater representation than other groups. This divergence from White- or European-biased representation is likely the result of targeted population studies in communities suffering from a higher prevalence of these diseases.[39–42]

In addition to utilizing FDA data, we also examined resources from the European Medicines Agency (EMA),[43] ClinicalTrials.gov,[44] and the World Health Organization (WHO).[45] The EMA provides data on clinical trials conducted within Europe, ClinicalTrials.gov aggregates information from clinical trials conducted worldwide, and the WHO International Clinical Trials Registry Platform[46] compiles data from various international registries. However, none of these resources offers summary statistics on participant demographics, including ancestry. EMA, ClinicalTrials.gov, and WHO require reviewing each study individually to determine whether demographic data are available, and even then, there is no assurance that such data will be included.

The absence of readily accessible demographic information for these studies poses a significant barrier to addressing and reducing health disparities across different ancestries. Researchers who seek to conduct demographic data analyses to track and monitor disparities in diversity and inclusion of the resources must extract and compile the data manually, which hinders efforts toward equitable representation in clinical trials. The generalizability of research findings thus continues to be limited across diverse populations, and it is often unclear to whom they are (and are not) applicable.

The lack of diverse representation leads to poorer health outcomes for patients from underrepresented ancestral backgrounds in many clinical use cases.[47] Examples include studies in which genetic variability in drug metabolizing enzymes are found to contribute to a high number of adverse drug reactions (ADRs) reported in Africa.[48,49] *CYP2D6*, a gene involved in the metabolism of up to 25% of the drugs that are in common use in the clinic,[50] offers a case in point. Three alleles in *CYP2D6* are associated with poor breast cancer outcomes for African patients treated with tamoxifen.[51]

Codeine, a common analgesic drug, is banned in Ethiopia due to its adverse effects associated with variants of *CYP2D6*.[52] This is attributed to a gene duplication that causes serious adverse outcomes in 30% of a local Ethiopian population following codeine administration.[52] Other studies have also reported variants in *CYP2D6* (prevalent among north Africans) with the potential for toxic effects of administering codeine.[53,54] This toxicity may be a consequence of ultrarapid metabolism mediated by the enzyme encoded by *CYP2D6*, as the use of codeine by ultrarapid metabolizers can result in a significantly increased risk of respiratory depression, fatal concentrations of morphine in breast milk, or even death.[54] Due to the presence of this common, highly penetrant pharmacogenetic variant in the absence of economic and logistical feasibility of genotyping the Ethiopian population, the total prohibition of codeine has been implemented.[52]

To assess disparities in the evidence for gene-drug interactions involving *CYP2D6*, we analyzed predicted metabolizer phenotypes assigned to known PGx alleles by PharmGKB[55] (Table S2) and constructed biogeographical group frequencies

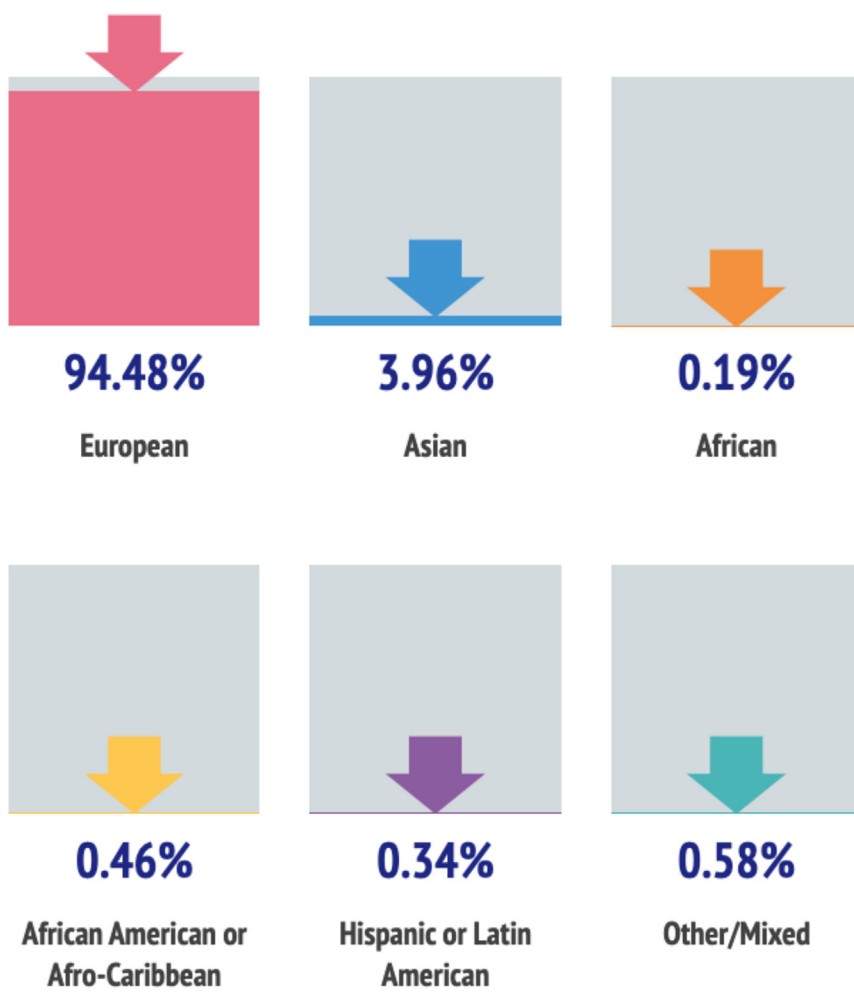

## Total GWAS participants diversity

Version 1.0.0. Last check for data: 2024-09-09 00:21:35 .

| 94.48% | 3.96% | 0.19% |
| European | Asian | African |

| 0.46% | 0.34% | 0.58% |
| African American or Afro-Caribbean | Hispanic or Latin American | Other/Mixed |

**Figure 3. Total proportion of participants in the GWAS Diversity Monitor**
The total proportion of representation from any population other than Asian (3.96%) or European (94.48%) is <1%, suggesting that current efforts toward diversity in genomics are failing. Source: GWAS Diversity Monitor (https://gwasdiversitymonitor.com/).

Warfarin is another commonly prescribed drug worldwide, which has been used in the treatment of cardiovascular disease for more than 60 years.[58] However, it is reported to be among the top four drugs leading to ADR-driven hospitalization in South Africa.[58] This also affects other parts of Sub-Saharan Africa.[59] Most studies of individuals with African ancestry using warfarin have been conducted in the United States and Brazil, which limits the generalizability of these findings to the development of precise dosage protocols in Sub-Saharan African populations.[60] Consequently, risk prediction for warfarin over-anticoagulation (estimated in 18%–24% of cases overall) is limited to individuals of (mostly) European ancestry, who exclusively benefit from precise evidence-based dosing protocols.[61]

European-biased evidence leading to exclusive translational healthcare benefits is unfortunately quite common. Genomic risk prediction models using GWAS discovery results from the UK Biobank are known to be less accurate when applied to non-European target populations, with Africans benefiting the least from these models.[13] The transferability of genetic models varies among African populations, with some benefitting from more precise genetic risk scores when using African American individuals as a reference.[62] However, individuals of many different ancestries and backgrounds benefit from better risk prediction models when the GWAS discovery data that seed these models include genomic diversity from African populations. It is therefore imperative to prioritize the inclusion of data from individuals of diverse recent African ancestral backgrounds for the benefit of all recipients of genomic medicine.

## EQUITY IN ACCESS TO GENETIC TESTING

The availability of direct-to-consumer (DTC) genetic testing has been fueled by companies like 23andMe, Ancestry.com, and MyHeritage, where genotyping can be performed at a cost that ranges from $100–$200 (USD). Although these costs are affordable to many customers in high-economy nations, they are prohibitively expensive for most people in low- to middle-income

according to allele activity[56] (Figure 6). We defined an ultrarapid metabolizer as one exhibiting a phenotype with an activity score >2.25, normal metabolizer 1.25–2.25, intermediate metabolizer 0.25–1, poor metabolizer 0, and indeterminate metabolizer as not applicable. Among Oceanians,[57] 18% were classified as having an ultrarapid metabolizer phenotype, which is a 14% excess compared to the global average of 4%.

There is also a disproportionate fraction of Sub-Saharan Africans (frequency = 0.35) with an indeterminate metabolizer status, while no other biogeographical group exceeds a frequency of 0.09. This excess of missingness in the form of "indeterminate metabolizer status" most likely reflects the genetic diversity in Sub-Saharan Africans (i.e., alleles previously unknown, with no predicted clinical phenotypes) that are missing from the PGx evidence base. This suggests there is less certainty in the safety and efficacy of drugs metabolized by *CYP2D6* for many African ancestries, regions, and populations.

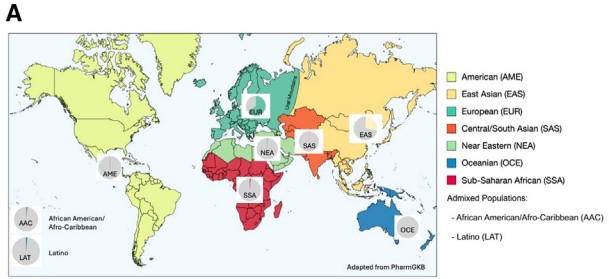

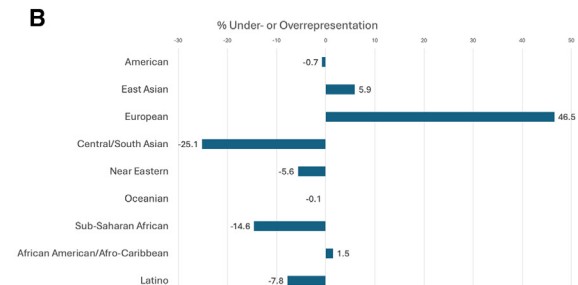

**Figure 4. Biased representation and missing global diversity in pharmacogenomics**

(A) Pie charts reflect the percentage of individuals included in PharmGKB-curated studies with respect to the total number of individuals. Europeans (EUR) make up 63.6%, 28.1% east Asian (EAS), 2.2% central/south Asian (SAS), 2.1% African American or Afro-Caribbean (AAC), 1.6% Sub-Saharan African (SSA), 1.6% Latino (LAT), 0.9% Near Eastern (NEA), 0.1% Indigenous American (AME), and 0% Oceanian (OCE).[31]

(B) Difference in percentage of ancestries between global census and representation in pharmacogenetic studies. A percentage of 0 represents a balanced proportion as compared to the share of the population globally. We note inconsistencies in labeling and coloring of populations due to different ways of reporting ancestries by sources. (Rough estimates of global biogeographical populations, including their diaspora, were calculated using sources available in Table S1.)

countries (LMIC). In the absence of widespread access to insurance coverage for clinical genetic testing in many countries and limited capacity for genetic testing in others, DTC genetic testing offers some genetic information to those who can afford to take advantage of these services. Although DTC genetic testing results may not be produced with the quality controls required of clinical testing, some argue they should be globally accessible regardless of utility.

Repositories such as openSNP[63] and the Personal Genomes Project[64] allow data donors to share genotype results from their own DTC tests, which then become available for public use on the repository websites. In an experiment carried out in 2017 by Shaw and Corpas,[65] 23andMe genotypes from open access data resources were used to evaluate sample diversity. After downloading and cleaning 3,137 genotype data files to remove duplicates and filter incomplete entries, they analyzed a dataset of 2,280 unique, individual files. Using principal-component analysis from 2,402 phase 3 1000 Genomes Project samples,[66] three continental clusters from the study (European, Asian, and African) were constructed using metrics of genetic distance; then, the curated genotype data from 2,280 DTC customers were projected into the principal-component space of 1000 Genomes Project data, allowing Shaw and Corpas to assign continental ancestries to individuals based on their 23andMe reported genotypes. Table 1 summarizes the predicted genetic ancestry proportions from curated DTC genotypes.

This analysis has some important limitations. First, it was performed in 2017. Since then, 23andMe has launched campaigns to recruit customers with more diverse sociocultural and ancestral backgrounds.[67] Second, the approach assumes no systematic biases due to differences in cultural values that might influence people's willingness to upload their personal genotype information from DTC tests to open, public repositories. It may be that biases observed in those who choose to leverage these third-party resources do not reflect the true nature of disparities in access to DTC genetic testing. Third, biogeographical region and continental-level ancestry assignment are poor proxies for genomic diversity; indeed, there is a rich genetic landscape across each continent, with more shared genomic variants in

common (between continents) than unique to one. Fourth, this study analyses only 23andMe data because the format they use is the only type available in the public resources used in this analysis.

Notwithstanding these limitations, we cross-referenced these numbers using 23andMe data with a more up to date statistic from the International HundredK+ Cohorts Consortium,[68] where 23andMe has a current enrollment of 10 million individuals. Only approximate figures are provided by the International HundredK+ Cohorts Consortium. According to these numbers, 23andMe's 10-million-person cohort consists of an ancestry that is 1%–25% Black, African American, or African ancestry; 51%–75% European; 1%–25% Latino or Spanish; and 1%–25% Middle Eastern or north African.

We also researched the information that 23andMe provides in their Research Innovation Collaborations Program[69] (Table S3). They suggest that race and ethnicity categories inferred from genetic data are highly correlated with self-reported race and ethnicity (but they are not always the same). They use genetic ancestry as a proxy for self-reported race and ethnicity, yielding the numbers below. While we do not endorse the use of race and ethnicity as satisfactory for describing diversity, we reuse 23andMe source data to report meaningful results for existing ancestries that have taken DTC tests.

## DISCUSSION

Historically, genomic research has predominantly focused on populations of European descent, producing genetic databases and biobanks rich in data from these populations. The funding and infrastructure systems in Europe and North America have facilitated the advancement of genomic technologies that benefit local populations, which have led to reference genomes and genetic markers being more tailored to European populations. In addition, healthcare systems from these countries allow better access to genomics as part of patient care, enabled by policies and regulations that support the use of genomics technology in healthcare. All these factors create a cumulative advantage for European ancestry populations. Concrete

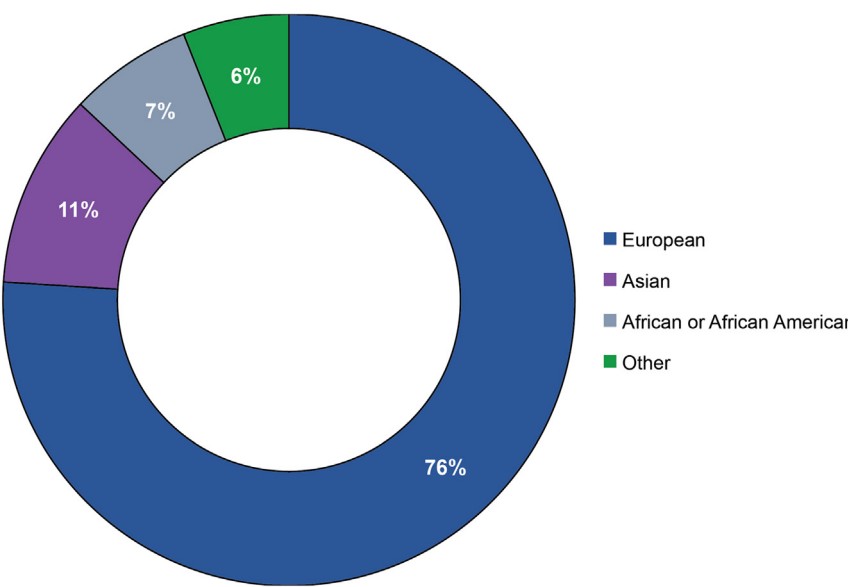

Percent of clinical trial participants by ancestry (2015 - 2019)

- ■ European
- ■ Asian
- ■ African or African American
- ■ Other

**Figure 5. Individuals taking part in clinical trials between 2015 and 2019 segmented by population categories reported by the FDA**
"Other" includes populations such as Latino or Oceanians, whose lack of data is particularly evident. We note inconsistencies in labeling and coloring of populations due to different ways of reporting ancestries by sources. Data adapted from the FDA drug trial snapshot.[37]

examples have been given for how the genomic evidence base is biased: in GWASs, clinical trials, PGx, and DTC genetic testing. The urgency to address these disparities is increasing, particularly now that rapid advances in AI may amplify biases contained in existing datasets and derived models. To address barriers to equitable representation in genomic data across the globe, hurdles that need to be overcome include the following:

(1) limited resources and time for meaningful engagement with underrepresented populations and diverse biogeographical regions;
(2) technical barriers involving models or annotations based on mainly European ancestral backgrounds, rendering current genomic medicine and emerging precision medicine less effective for more diverse populations;
(3) lack of standards or metrics for measuring and reporting genomic diversity; and
(4) no clear targets or thresholds for achieving sufficient diversity and equity across organizations, institutions, and global initiatives.

While thresholds for appropriate inclusion and diversity in global genomic datasets remain elusive, current approaches for measuring diversity continue to be imprecise. This lack of precision for global diversity targets may also reflect poor choices for the classification of diverse groups, making comparability between groups among different data sources challenging.

### Limited available data and inconsistencies in population labels

A key challenge in our analysis is the inherent variability in how different datasets define and categorize populations. This variability arises from the use of multiple resources and tools, each with their own population labels, ancestry classifications, and country groupings. This poses significant barriers to achieving complete cohesion in our analysis and presentation of results.

The genomics databases we analyzed, including GWAS, PGx, DTC genetic testing, and FDA drug trials, define populations based on different criteria. For instance, the GWAS Diversity Monitor groups individuals broadly using categories such as European, Asian, African, African American or Afro-Caribbean, Hispanic or Latin American, and Other/Mixed. Other resources such as PharmGKB further divide populations into more specific subgroups such as east Asian, central/south Asian, Near Eastern, or Sub-Saharan African. Similarly, the terms "ancestry," "descent," and "ethnicity" are used interchangeably in some studies but defined more narrowly in others, adding confusion and variability.

Figures 2 and 3, derived from the GWAS Diversity Monitor, refer to broader geographic categories such as Asia. Figure 4A shows labels as east Asian, central/south Asian, and Near Eastern, reflective of the different classification system used by PharmGKB. The FDA drug trial snapshot reports differently populations of African origin, including under the same label African and African American. These differences are not arbitrary and reflect the underlying methodologies of the original datasets. As we strive to present a unified analysis, it is not always possible to align these labels across the paper without oversimplifying or misrepresenting the source data.

These challenges also extend to visual representations. We note that European is represented as pink by the GWAS Diversity Monitor (Figures 2 and 3), while PharmGKB represents European as green (Figure 4A). We recognize that this creates a disjointed appearance where the preservation of original color schemes and groupings are necessary to maintain the integrity of the sources.

It is important to note that some data sources limit their representation to specific populations, leading to underrepresentation of certain regions or ancestries. For instance, the term "Oceanian" appears only in PharmGKB and it is absent from the GWAS Diversity Monitor, DTC genetic testing, and ClinicalTrials.gov. This shortcoming is severe for the incumbent population, as it might skew the analysis toward regions or populations where genomic data are more readily available. It is therefore important to acknowledge ascertainment bias in some data sources we rely on, which significantly complicates the task of fully harmonizing a global view of genetic diversity. Although these

**Cell Genomics**
**Perspective**

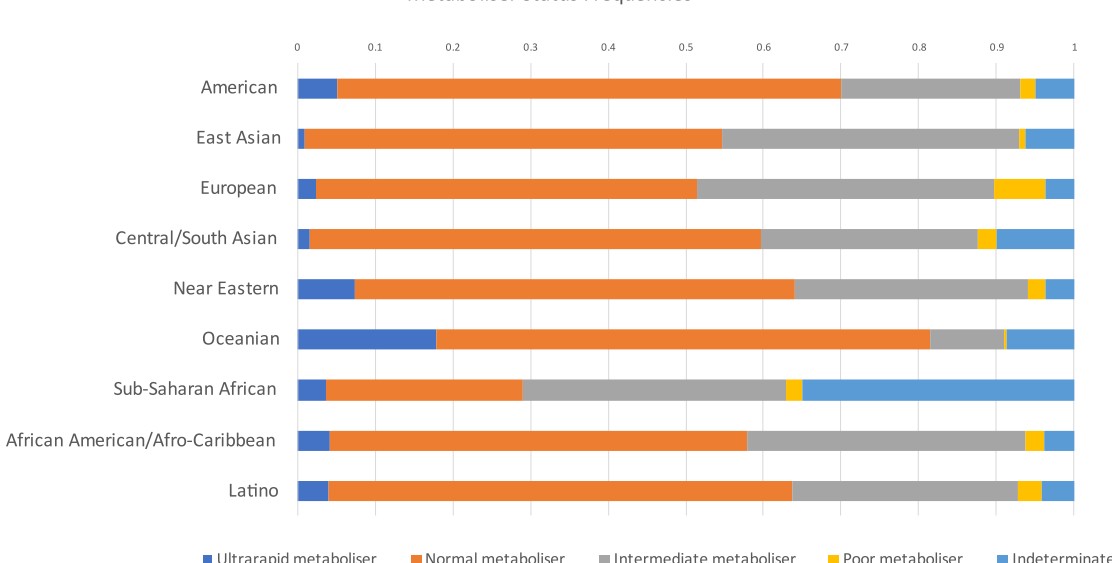

**Figure 6. PharmGKB predicted metabolizer phenotype frequencies for *CYP2D6*, according to biogeographical groupings used in the resource**
These data were adapted to reflect definitions of allele activity,[56] where ultrarapid metabolizer has an activity score >2.25, normal metabolizer 1.25–2.25, intermediate metabolizer 0.25–1, poor metabolizer 0, and indeterminant metabolizer not applicable.

challenges do not undermine the validity of our analysis, they highlight the need for greater diversity awareness and standardization of mainstream health genomic datasets.

### Increase of GWAS samples of Europeans driven by biobanks

A key distinction in genomic studies arises from different approaches taken by biobank-driven GWASs and those conducted by disease-focused consortia. A major limitation of biobank-driven GWASs is the potential overrepresentation of certain ancestral groups such as Europeans. This can skew findings toward this population.[14] Such overrepresentation can lead to double counting of the same samples across hundreds or even thousands of GWASs due to these datasets being used repeatedly across many studies.

Disease-focused consortia gather data from individuals affected by specific conditions, often including severe diseases not well represented in biobanks. These studies tend to involve smaller sample sizes due to the rarity of the diseases being studied, offering more targeted insights into the conditions. Disease-focused consortia may include more diverse populations, especially if they are related to conditions more prevalent in underrepresented groups.[12,34] Their smaller sample sizes and narrower focus, however, can limit their generalizability. To address these issues, future research will require both population-based and disease-focused consortia. The integration of both approaches will improve global representation in genome research.

### Standard metrics and targets for diversity

Balancing the proportions of populations represented in datasets based on fractions of the global census population is an un-

satisfactory metric of diversity and inclusion. New metrics are therefore needed for the scientific community to measure and identify the representation that has yet to be included in global data resources. Current approaches vary across contexts and resources; thus, standardization must also be considered. As mentioned above, the GWAS Diversity Monitor, PharmGKB, and the FDA have different criteria to select, assign, or categorize participants by genetic ancestry.

Proportional representation based on global census population ignores the potential for underrepresented populations to contribute previously unknown genomic variants. Sub-Saharan Africa has the most diverse genomic landscape globally, with many ancient and modern combinations of genetic ancestries.[70] As such, there should be more individuals from these parts of the world included in genomic studies and resources to adequately represent the human genetic diversity that they can contribute to the genomic evidence base.

Applied to the field of PGx, understudied populations with more diverse haplotype frequencies are more likely to be affected by imprecision in evidence-based guidance for drug dosage administration.[31] For instance, indeterminate metabolizer status (unknown clinical phenotype) based on variants of a gene that metabolizes 25% of prescribed drugs (*CYP2D6*) disproportionally affects Sub-Saharan Africans, suggesting that a number of alleles common in this biogeographical region are missing. In contrast, Europeans and east Asians (e.g., China, Japan, South Korea), are overrepresented in PGx datasets relative to their share of the global population.

Using a global census population size to motivate proportional sampling and representation in genomic databases and biosample repositories also disadvantages smaller populations,

**Table 1. Breakdown of predicted continental ancestries from openly shared 23andMe genotypes**

| Predicted genetic ancestries | No. of unique individuals |
| --- | --- |
| African | 50 (2.2%) |
| Asian | 66 (2.9%) |
| European | 2,164 (94.9%) |
| Total | 2,280 (100%) |

who may also have distinct concerns or needs to be engaged and included. Indigenous Americans, comprising about 62 million individuals (according to global census estimates), is a much smaller population than the 2 billion central/south Asians, for example. The underrepresentation of south Asians in PGx datasets relative to their census population size is greater than those of Indigenous Americans or any other biogeographical group. Importantly, there is no reported representation of Oceanians in PharmGKB, the GWAS Diversity Monitor, DTC genetic testing, or ClinicalTrials.gov. For all groups that have low numbers worldwide, there are likely historical reasons for their relative population sizes being smaller than others, for example, because of attempted genocide or colonization. As such, it is critical not to exclude these groups from genetics and genomics research, especially based on a justification that there are so few of them across the globe.

### Overcoming genetic colonialism

The urgent need for an increase in diverse genomic data also extends to populations who have suffered the consequences of genetic colonialism.[71] Genetic colonialism refers to the exploitation of research participants from marginalized communities, where researchers have often failed to be fully transparent about their research intentions or the outcomes. This is exemplified by practices that exploited research participants by not being completely open about research intentions or outcomes.[70] These unethical practices have not only eroded trust but have also led to the misappropriation of genetic resources and data. Addressing this issue is crucial for ensuring ethical research practices and for promoting Indigenous data sovereignty in particularly vulnerable regions such as Latin America or Australia, which advocates for the rights of Indigenous peoples to control their own genetic and genomic information.[72] Colonialism, at its core, does not center respectful engagement with people labeled "other." This may have influenced Western scientists to treat potential research participants in communities that are foreign to them with little regard for autonomy, respect for persons, benefit sharing, informed consent, or any of the other principles and practices that are central to bioethics. There has been substantial harm done through research relations with Indigenous and local communities, resulting in mistrust and unwillingness to participate or contribute.[73,74] Therefore, respectful and reciprocal approaches are needed to engage with diverse populations and communities.[25,47]

Ongoing efforts to address the impacts of colonialism on genetics/genomics research include the development and applications of CARE Principles for Indigenous Data Governance.[75,76] These principles can be seen as complementary to FAIR (find-

ability, accessibility, interoperability, and reusability) principles for open data sharing.[77] Further efforts may succeed in drawing on the United Nations Declaration on the Rights of Indigenous Peoples, which reaffirms the rights of Indigenous peoples to control data about their peoples, lands, and resources. The colonialist (and eugenics-laden) history of genetics as a field cannot be undone, but analytic approaches, data/sample governance models, and engagement practices can be developed and implemented to chip away at the harmful effects of our past.

### Increasing access to data and technology

If we are to expand the benefits of human genomics to all peoples, DTC genetic testing products and services have a role to play. First, DTC genetic tests make it easier for individuals to access their genetic information without the need for a healthcare provider or a medical prescription. This democratizes access to personal genetic data, allowing people from various backgrounds to learn about their ancestral origins and potential health risks (although the latter are contended and very limited). Second, by making genetic testing more widely available, DTC companies could also play a role in increasing public awareness and knowledge about genomics. This can stimulate interest in personal and family health histories.

In several countries, such as Germany, France, and Italy, strict regulations on DTC genetic testing limit its availability due to concerns about privacy, misinterpretation, and the absence of medical guidance.[78,79] These regulations are designed to protect consumers but reduce access compared to regions with more lenient laws like the United States. However, it is important to distinguish this issue from the broader lack of diversity in genomic research, which remains a significant challenge across large-scale studies.

Disparities in access to DTC genetic testing are paralleled by biased models of genetic risks and reports, which are tailored to European ancestries and norms, including in the interpretation, reporting, and communication of results.[80] Although some cultural inclusion efforts are now under way,[81] when it comes to technology, most of the genotype markers and the bulk of annotations in genomic datasets are still based on individuals of mostly European descent.[82]

### CALL TO ACTION

It is imperative to acknowledge the limitations of applying a Western (European-centric) perspective on healthcare to global initiatives, as this may not align with the values and preferences of different populations. To ensure that health interventions are effective and culturally appropriate, cultural humility is needed, to respect and integrate local preferences, needs, and paradigms. What is beneficial in one context might be seen as intrusive or problematic in another. This highlights the necessity of partnering with local communities to understand their specific needs, values, and desires. Such an approach not only ensures cultural relevance but also enhances the acceptance and sustainability of health initiatives. Thus, by respecting and recognizing the rich diversity of cultural perspectives on health and well-being, we can foster more equitable approaches to conducting research and developing biomedical resources.

Article 15 of the International Covenant on Economic, Social and Cultural Rights[83] is an international human rights treaty adopted by the United Nations in 1966[84] that requires states to recognize the right of everyone to enjoy the benefits of scientific progress and its applications. It also stipulates that these benefits shall be enjoyed while respecting the freedom to develop scientific research and recognizing that international cooperation in the sciences benefits all. Similarly, United Nations Educational, Scientific and Cultural Organization's Universal Declaration on the Human Genome and Human Rights, states that "everyone has a right to respect for their dignity and for their rights regardless of their genetic characteristics" and to respect their uniqueness and diversity.[85] It is therefore by invoking these treaties that we call upon international research organizations and leaders to enhance investments in capacity building and infrastructure and/or accessible genomic testing for underrepresented populations.

While great strides have been made to expand human genetics and genomics globally through international initiatives such as H3Africa,[86] the Latin American Genomics Consortium,[87] and the Equity, Diversity, and Inclusion Advisory Group for the Global Alliance for Genomics and Health,[88] efforts to date remain insufficient for data equity.

Concurrently, academic and industry partnerships are needed that respect the research needs of the Global South. To date, many of these partnerships involve researchers in LMICs being mentored by colleagues abroad on conditions that may result in a greater emphasis on Eurocentric and US-focused research interests.[89] It is possible that this model further widens the gap between nations, breeding distrust and resentment. As such, it is essential for everyone involved to be aware of historical and current power dynamics, including differential incomes and wealth. Truly equitable partnerships require a reconciliation of these dynamics through active effort.

We recognize that environmental, cultural, and socioeconomic factors are integral to fully understanding human diversity. Future research should seek to integrate these broader cultural elements for a more holistic approach to understanding diversity within precision medicine and healthcare equity.

## CONCLUSION

Despite efforts to diversify genomic databases, data from GWASs, PGx, clinical trials, and DTC genetic testing lack equitable global representation. Most genomic data in the public domain are from individuals of European descent, with alarmingly scant inclusion of other ancestries, particularly Sub-Saharan African, Indigenous American, and Oceanian. This bias undermines the universal utility of genomic medicine while perpetuating healthcare disparities.

The persistence of ancestral biases in GWASs, despite accessible diversity metrics and real-time monitoring, indicates that the current strategies for inclusion are insufficient. These biases extend beyond GWASs, as seen in PGx databases and clinical trial demographics, with tangible consequences for equity in drug efficacy and safety. In the absence of reliable evidence, there are increased risks for underrepresented populations, as exemplified by the gene-drug interactions of enzymes like *CYP2D6*. Underrepresentation is further evidenced in the context of DTC genetic testing, where the participation of non-European ancestries remains nominal.

Advancement toward a more equitable genomic landscape will require standard metrics and clear, consistent targets for diversity. Additionally, combating genetic colonialism and increasing access to testing services are essential steps toward more inclusive genomics. Our urgent call to action invokes international human rights treaties, emphasizing the right of everyone to benefit from scientific progress, which includes access to genomics. International research organizations, industry leaders, and policymakers can foster investments to support the development of resources and results that reflect global human genomic diversity. Only then can the promise of precision medicine be realized for all individuals, regardless of their national origin or ancestral background.

### ACKNOWLEDGMENTS

We are grateful to Kelly Ormond and Effy Vayena for insightful comments on early versions of the manuscript. We would like to thank Vicente Soriano for useful comments on revisions of the manuscript.

### AUTHOR CONTRIBUTIONS

M.C. designed the study, performed the analyses, and wrote the paper. M. Pius performed analysis on clinical trials. M. Poburennaya helped design the figures and contributed manuscript edits. H.G., M.D., S.N., C.L.-C., A.P., and S.F. helped in the design of the study, provided expert advice, and contributed edits to the manuscript.

### DECLARATION OF INTERESTS

M.C. is a founder of Cambridge Precision Medicine Limited and a member of its scientific advisory board.

### SUPPLEMENTAL INFORMATION

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
