## [Document S2. Transparent peer review records for Corpas et al · Cell Genomics]

Summary

Initial submission: Received : 3/26/2024

Scientific editor: Laura Zahn

First round of review: Number of reviewers: 3
Revision invited : 5/13/2024
Revision received : 7/13/2024

Second round of review: Number of reviewers: 3

Third round of review: Number of reviewers: 2
Accepted : 11/19/2024

Data freely available: Yes

Code freely available: Yes

This transparent peer review record is not systematically proofread, type-set, or edited. Special characters, formatting, and equations may fail to render properly. Standard procedural text within the editor's letters has been deleted for the sake of brevity, but all official correspondence specific to the manuscript has been preserved.

Referees' reports, first round of review

Reviewer #1: Corpas et al. present their perspective on how biomedical research continues to fail to achieve ancestral and biogeographical diversity. They particularly focus on GWAS, pharmacogenomics, clinical trials, and direct to consumer genetic testing.

Underrepresentation is an ongoing challenge, that needs ongoing recognition cognizant efforts to improve. However, I think there are several gaping holes in this piece.

1. Recognition that past misconduct is a major source of distrust in the scientific research community should occur in the introduction.
2. This makes it appear GWAS are doing worse, when in fact the absolute number of non-European participants in GWAS studies is increasing. The GWAS Diversity Monitor counts a participant for each GWAS they are in. As a result, in 2021 there were 3675.9 million participants in the United Kingdom, which has a population of ~67.5 million. This level of deep phenotyping where an individual would be included for many traits/phenotypes is more likely to occur when genetic data is linked to EHR data.
3. It should be acknowledged that 2021 contains a plethora of publications based on the UK biobank data, which is generally representative of the country's population.
4. The increased proportion of studies with "missing" ancestry information, needs to be recognized as a sign of progress. This is not captured in trans-ancestral studies and may also reflect recognition that race/ethnicity is not the same as ancestry. It should also be acknowledge that skipping ancestry/population specific GWAS may results in Whitewashing results.
5. It would be useful to note progress that has been made.
6. Using US FDA data to assess diversity in clinical trials is unlikely to adequately capture trials run outside of the US. What other trial monitoring systems are available?
7. Numbers of 23andMe consumers enrolled in research should be published information. I would assume higher participation rates from European ancestry customers.
8. What are barriers to establishing or expanding DTC companies for besides expense of the test to consumer? Are there regulatory/legal barriers? Consumer demand?
9. This plea makes the assumption that a Western approach to health care is the desirable for all populations. Recognition of different values is important. What we may perceive as benefitting all, may be viewed as an unwelcome intrusion and introduction of problems to other societies. I recognized this is a fine needle to thread.

Reviewer #2: Dear Editor,

While this Perspective addresses an important topic, I'm concerned with its novelty and technical quality. Many analyses in this study appear to have been designed for convenience, were sloppy, and may contain factual errors (e.g., using GPT to estimate a population size) or citing data sources unable to be verified easily (e.g., DTC research report, \$4900 single user access, not covered by most journal subscriptions).

Reviewer #3: Comments enter in this field will be shared with the author; your identity will remain anonymous.

I am glad to see a quantitative, multi-aspect, approach to documentary the lack of diversity in human genetic studies. The choices of CYP2D6 and of warfarin as examples are particularly well-chosen.

My comments are limited to minor suggestions for improvement.

* The title is a little misleading. I don't think the paper really address "why we're still failing in diversity"

* Although it's not my place to criticize the GWAS Diversity Monitor (shown in Figure 1), it's a little ironic that the projection used by the monitor overemphasizes the area of N America, Europe, and Russia.

* The last sentence ("However, the last ...") in the middle paragraph of page 3 is confusingly written ("greater proportions of missing ancestries . . . with as few as")

* Figure 3A could do with some more explanation. For example, this is the first use of the "American" label, which readers might need some help in recognizing as indigenous American. The super-imposed pie charts are hardly self-explanatory either.

* Page 8, first paragraph, there appears to be a gap (between 1 and 1.25) between intermediate metabolizer and normal metabolizer

* Page 11, last paragraph. The first sentence is a bit hard to understand, and if I am guessing correctly as to what it means, it is contradicted by the next sentence.

Authors' response to the first round of review

We would like to thank Reviewer 1, 2 and 3 for suggested amendments and comments. In what follows we address each of their comments. Please note that we have not kept all edits we have made to our new version in tracked changes (otherwise the document would be cumbersome to read). Hopefully it can be seen how substantially the manuscript has evolved based on their feedback in order to address all reviewer comments to their satisfaction.

Reviewer #1:

Corpas et al. present their perspective on how biomedical research continues to fail to achieve ancestral and biogeographical diversity. They particularly focus on GWAS, pharmacogenomics, clinical trials, and direct to consumer genetic testing.

Underrepresentation is an ongoing challenge,

that needs ongoing recognition cognizant efforts to improve. However, I think there are several gaping holes in this piece.

1. Recognition that past misconduct is a major source of distrust in the scientific research community should occur in the introduction.

We acknowledge this suggestion and have added the following paragraph in the background section (lines 51-59), which suggests two historical instances of exploitation, the Tuskegee Syphilis Study and

the unauthorised use of Henrietta Lacks' cells:

“While achieving diverse representation in datasets is critical for human health, efforts to diversify participation in genomic research face significant challenges, including a deep-rooted mistrust in the scientific community. This mistrust often stems from past misconduct and unethical practices in research, particularly involving marginalised communities. Historical instances of exploitation, such as the Tuskegee Syphilis Study ³ and the unauthorised use of Henrietta Lacks' cells ^{4,5}, have left a legacy of scepticism and wariness towards scientific research among underrepresented populations. Acknowledging and addressing these historical injustices is crucial for rebuilding trust and fostering greater participation from historically underserved populations.”

2. This makes it appear GWAS are doing worse, when in fact the absolute number of non-European participants in GWAS studies is increasing. The GWAS Diversity Monitor counts a participant for each GWAS they are in. As a result, in 2021 there were 3675.9 million participants in the United Kingdom, which has a population of ~67.5 million. This level of deep phenotyping where an individual would be included for many traits/phenotypes is more likely to occur when genetic data is linked to EHR data.

We thank Reviewer #1 for bringing this to our attention. Because Each individual is counted for every study they are part of, the GWAS Diversity Monitor can report a participant count which exceeds the actual population size. Consequently, the figure of 3675.9 million participants reflects repeated counts of the same individuals across different traits and phenotypes.

We have added the following paragraphs to account for this point (lines 195-201):

“Despite its usefulness, the GWAS Diversity Monitor may report a participant count that can exceed the actual population, due to its methodology. Since each individual is counted in every study they are part of, in 2021 the GWAS Diversity Monitor showed 3675.9 million participants in the United Kingdom, which has a population of 67.0 million. This reflects repeated counts of the same individuals across different traits and phenotypes. Such double counting may make diversity worse than it is, as the absolute number of diverse genomes is increasing ²².”

3. It should be acknowledged that 2021 contains a plethora of publications based on the UK biobank data, which is generally representative of the country's population.

We have added this text between line 106 and 122:

“The predominance of GWAS publications using the ‘White/British’ samples from the UK Biobank should therefore be considered when interpreting participant numbers and the representation of genetic diversity in GWAS.

It is also important to note that UK Biobank data include >20k individuals with group labels other than ‘White/British’ which are regularly excluded from genetic analyses using this dataset but could be a useful resource for contributing GWAS results from more diverse genetic ancestral backgrounds. The widespread reliance on ‘White/British’ UK Biobank data for GWAS highlights both the strengths and limitations of using such a centralised and comprehensive dataset. While it enables detailed and extensive genetic research by linking electronic health records (EHRs) to genetic data, it also underscores the need for diverse

representation in genomic studies to ensure findings are applicable to broader populations. Understanding that there may be more to discover in UK Biobank subsamples from other ethnic or ancestral groups than by repeated GWAS for the same traits using 'White/British' samples may yet motivate researchers to conduct analyses with smaller sample sizes, but more predictive power."

4. The increased proportion of studies with "missing" ancestry information, needs to be recognized as a sign of progress. This is not captured in trans-ancestral studies and may also reflect recognition that race/ethnicity is not the same as ancestry. It should also be acknowledged that skipping ancestry/population specific GWAS may results in Whitewashing results.

Thanks for the above comment. To address it, we have added text that acknowledges studies with "missing" ancestry information as a sign of progress, discussing the potential consequences of skipping ancestry-specific GWAS (lines 127-134):

"Recent years have seen an increased proportion of GWAS reporting "missing" ancestry information. This trend should be recognised as a sign of increasing precision and transparency in human genetics research. It reflects a growing understanding that race and ethnicity are distinct from genetic ancestry 15, which is crucial for accurate data interpretation and representation. We have also seen progress towards more inclusive studies that combine and transcend broad ancestral groupings, moving beyond simplistic racial categorisations to a more nuanced understanding of genetic diversity 16,17."

5. It would be useful to note progress that has been made.

We agree and have added a section highlighting progress being made with African Genomes (lines 134-138):

"Furthermore, initiatives such as the African Genome Variation Project 18 and the H3Africa Consortium 19 have significantly expanded the repertoire of available genomic data from modern African populations 20. These projects aim to understand genetic diversity in different African populations, and its implications for human health and disease; thus increasing representation of diverse African ancestries in research."

6. Using US FDA data to assess diversity in clinical trials is unlikely to adequately capture trials run outside of the US. What other trial monitoring systems are available?

Thanks for this insightful comment. We checked a) the European Medicines Agency (EMA), which contains data pertaining Europe, b) ClinicalTrials.gov, which is a collection of worldwide clinical trials and c) the World Health Organisation (WHO). None of these resources provide summary statistics on ancestry information. All require going through each study independently to see if any demographics data are available, with no guarantee that it exists. We believe that this is in itself a hurdle that contributes to the detriment health equity across different ancestries. We have added the following text as a result (lines 296-304):

"In addition to utilising FDA data, we also examined resources from the European Medicines Agency (EMA) 37, ClinicalTrials.gov 38, and the World Health Organization (WHO) 39. The EMA provides data on clinical trials conducted within Europe, ClinicalTrials.gov aggregates information from clinical trials conducted worldwide, and the WHO International Clinical Trials Registry Platform (ICTRP) 40 compiles data from various international registries.

However, none of these resources offer summary statistics on participant demographics, including ancestry. EMA, ClinicalTrials.gov and WHO require reviewing each study individually to determine if demographic data are available, and even then, there is no assurance that such data will be included.”

7. Numbers of 23andMe consumers enrolled in research should be published information. I would assume higher participation rates from European ancestry customers.

Thanks for the suggestion. We have been able to do some further research and were able to find summary statistics on 23andMe’s website that publish summary statistics on the diversity of 23andMe research cohorts (<https://research.23andme.com/research-innovation-collaborations/>). We have added this information to the paper (lines 437-449):

“We also researched the information that 23andMe provides in their Research Innovation Collaborations Program 63 (Table 2). They suggest that race and ethnicity categories inferred from genetic data are highly correlated with self-reported race and ethnicity [but they are not always the same]. They use genetic ancestry as a proxy for self-reported race and ethnicity, yielding the numbers below.

Predicted genetic ancestries	Approximate number of participants
Latino	1,312,000
African American	443,000
East Asian	314,000
West Asian and North African	82,000
South Asian	83,000
Other non-European	365,000

Table 2. 23andMe’s inference of genetic ancestry of participants within the ~10 million that have consented for research.

Based on the publicly available data we found to characterise inclusion/access to or use of DTC genetic testing, the European bias seen in research datasets is also seen in the DTC landscape.”

8. What are barriers to establishing or expanding DTC companies for besides expense of the test to consumer? Are there regulatory/legal barriers? Consumer demand?

Thanks for the comment. We have added a paragraph to the discussion on “Increasing Access to Data and Technology” (lines 601-611). This paragraph states the following:

“The market for DTC genetic testing services also still mostly focused on American and European markets, excluding large swaths of the globe, especially Africa and Latin America. This lack of access to technology is further exacerbated by the cost of USD \$100-\$200 for a DTC genetic test, which is not affordable to most people in low- and middle-income countries (LMICs). Establishing or expanding DTC genetic testing companies in regions outside of North America and the European Union involves navigating a complex set of challenges. One of these challenges is the lack of clear regulatory frameworks. Building relationships with local health authorities can help companies stay ahead of regulatory changes and ensure compliance. Additionally, collaborating with local legal experts who understand the intricacies of regional laws can help navigate legal barriers more effectively.”

9. This plea makes the assumption that a Western approach to health care is the desirable for all populations. Recognition of different values is important. What we may perceive as

benefitting all, may be viewed as an unwelcome intrusion and introduction of problems to other societies. I recognized this is a fine needle to thread.

We appreciate the reviewer's insightful comment regarding the assumption that a Western approach to healthcare is universally desirable. We fully recognise the importance of acknowledging and respecting different cultural values and healthcare paradigms. Our intention is not to impose a Western model of healthcare on all populations but rather to highlight the potential benefits of certain health interventions that have shown efficacy in diverse contexts. We have added the following paragraph in our call to action section to acknowledge the assumptions of Western health approaches being universally acceptable to all populations (line 621-631):

"It is imperative to acknowledge the limitations of applying a Western (European colonialist) perspective on healthcare to global initiatives, as this may not align with values and preferences of different populations. To ensure that health interventions are effective and culturally appropriate, cultural humility is needed, to respect and integrate local preferences, needs, and paradigms. What is beneficial in one context might be seen as intrusive or problematic in another. This highlights the necessity of partnering with local communities to understand their specific needs, values and desires. Such an approach not only ensures cultural relevance but also enhances the acceptance and sustainability of health initiatives. Thus, by respecting and recognising the rich diversity of cultural perspectives on health and well-being, we can foster more equitable approaches to conducting research and developing biomedical resources."

Reviewer #2:

While this Perspective addresses an important topic, I'm concerned with its novelty and technical quality. Many analyses in this study appear to have been designed for convenience, were sloppy, and may contain factual errors (e.g., using GPT to estimate a population size) or citing data sources unable to be verified easily (e.g., DTC research report, \$4900 single user access, not covered by most journal subscriptions).

We appreciate the reviewer's comment on the novelty and technical quality of our Perspective. Addressing these concerns is crucial for ensuring the rigour and reliability of our analyses. Here, we outline our responses to the specific points raised:

1. Novelty and Technical Quality: To our knowledge, existing literature has focused on analysing underrepresentation of one type of data source, for instance, polygenic risk scores (PRS) (Martin et al), GWAS (Fatumo et al) or Pharmacogenomics (Corpas et al). Some of the published analyses are now 5 years old and all of them are disjointed by being separately considered. The novelty of our study lies in the integration of these analyses with up-to-date numbers. This also allow us to comment in progress made since the initial publications.

An integrated assessment of genetic diversity across global datasets also allows us to appreciate current shortcomings in addressing global data equity disparities and how the current lack of progress is having tangible detrimental effects for some global populations. For instance, we provide a novel analysis on the CYP2D6 pharmacogene, which shows how a remarkable difference exists for African individuals who disproportionately suffer the presence of indeterminate metaboliser status compared to other genetic ancestries.

2. Analyses Designed for Convenience: We recognise the importance of rigorous study design and agree that convenience should not compromise the integrity of our analyses. Moving forward, we have ensured that all methodologies are thoroughly vetted and align with best practices in the field. Specifically, we have revisited our data collection and analysis methods to consider any potential shortcomings.

3. Use of GPT for Estimating Population Size: Our approach of using GPT has now been replaced throughout with census data or demographic studies as follows. We have added the table below to our supplementary materials for reference to our readers. We have changed Figure 4B to take population estimates according to the ones shown on the table below.

Region	Census	Global % (8bn)	Year	Source
American	62,000,000	0.80%	2024	https://en.wikipedia.org/wiki/Indigenous_peoples_of_the_Americas
East Asian	1,661,506,318	20.8%	2024	https://www.worldometers.info/world-population/eastern-asia-population/
European	1,280,000,000	16.0%	2019	https://www.nature.com/articles/s41588-019-0394-y
Central/South Asian	2,047,895,034	25.6%	2024	https://www.worldometers.info/world-population/southern-asia-population/#:~:text=The%20current%20population%20of%20Southern,of%20the%20total%20world%20population.
Near Eastern	493,000,000	6.2%	2022	https://en.wikipedia.org/wiki/Demographics_of_the_Middle_East_and_North_Africa#:~:text=The%20demographics%20of%20the%20Middle,population%20was%20around%20493%20million.

Oceanians	6,500,000	0.1%	2010	https://www.culturalsurvival.org/publications/cultural-survival-quarterly/oceania-islands-land-people#:~:text=Due%20to%20colonial%20neglect%20and,cultural%20traditions%20and%20ecological%20adaptations.
Sub Saharan African	1,210,000,000	15.1%	2022	https://www.statista.com/statistics/805605/total-population-sub-saharan-africa/#:~:text=Total%20population%20in%20Sub%2DSaharan%20Africa%202022&text=This%20statistic%20shows%20the%20total,to%20approximately%201.21%20billion%20inhabitants.
Afro-American / Afro Caribbean	47,000,000	0.6%	2024	https://en.wikipedia.org/wiki/African_diaspora_in_the_Americas
Latino*	701,548,639	8.8%	2024	https://www.pewresearch.org/race-and-ethnicity/fact-sheet/latinos-in-the-us-fact-sheet/ https://worldpopulationreview.com/country-rankings/latino-countries

* Summed up populations of Latin America + Latino population of US

** There is some inevitable overlap between American and Latino populations

4. Inaccessible Reference: There is a data source that is collected from the reference “DTC research report, \$4900 single user access” [<https://www.precedenceresearch.com/direct-to-consumer-genetic-testing-market>] [reference #64]. The numbers that we took are available from the summary page, for which it is not necessary to pay to have access.

Reviewer #3:

I am glad to see a quantitative, multi-aspect, approach to documentary the lack of diversity in human genetic studies. The choices of CYP2D6 and of warfarin as examples are particularly well-chosen.

My comments are limited to minor suggestions for improvement.

* The title is a little misleading. I don't think the paper really address "why we're still failing in diversity"

We have revised the title to “Genomics Still Failing on Diversity”.

* Although it's not my place to criticize the GWAS Diversity Monitor (shown in Figure 1), it's a little ironic that the projection used by the monitor overemphasizes the area of N America, Europe, and Russia.

This unfortunately is outside of our control. We have nevertheless noted in lines 176-177 that the projection map may overemphasizes the area of North America, Europe, and Russia.

* The last sentence ("However, the last ...") in the middle paragraph of page 3 is confusingly written ("greater proportions of missing ancestries . . . with as few as")

We have deleted this sentence as we thought it was too complicated to understand.

* Figure 3A could do with some more explanation. For example, this is the first use of the "American" label, which readers might need some help in recognizing as indigenous American. The super-imposed pie charts are hardly self-explanatory either.

Thanks for the suggestion. We have revised the legend for (now) figure 4A to the following (lines 254-259):

“Pie charts reflect percentage of individuals included in PharmGKB-curated studies with respect to the total number of individuals. 63.6% are European (EUR), 28.1% East Asian (EAS), 2.2% Central/South Asian (SAS), 2.1% African American or Afro-Caribbean (AAC), 1.6% Sub-Saharan African (SSA), 1.6% Latino (LAT), 0.9% Near Eastern (NEA), 0.1% Indigenous American (AME) and 0% Oceanian (OCE) [data taken from 30].”

* Page 8, first paragraph, there appears to be a gap (between 1 and 1.25) between intermediate metabolizer and normal metabolizer

We appreciate the reviewer's eye for detail regarding the apparent gap between intermediate and normal metaboliser phenotypes in our study, specifically within the range of 1 to 1.25. This gap is an artifact of the annotation system used for metaboliser status classification. The pharmacogenetic guidelines typically provide broad categories for metaboliser phenotypes based on activity scores or enzyme activity measurements. However, the boundaries between these categories can sometimes create artificial distinctions when applied to a continuous spectrum of metabolic activity. The gap observed in our data reflects the inherent limitations and discretisation of the annotation system rather than a true absence of intermediate metabolisers in this range. This issue highlights the challenges in

categorising a continuous variable into discrete phenotypic groups and underscores the need for careful interpretation of these classifications in pharmacogenetic research.

* Page 11, last paragraph. The first sentence is a bit hard to understand, and if I am guessing correctly as to what it means, it is contradicted by the next sentence.

Thanks for picking up the lack of clarity in this paragraph. We have deleted the last sentence that seemed to contradict the rest of the paragraph and rewritten the whole paragraph to make it clearer as follows (lines 543-555):

“Using a global census population size to motivate proportional sampling and representation in genomic databases and biosample repositories also disadvantages smaller populations, who may also have distinct concerns or needs to be engaged and included respectfully. Indigenous Americans, comprising about 62 million individuals (according to global census estimates), is a much smaller ‘population’ than the 2 billion Central/South Asians, for example. The underrepresentation of South Asians in PGx datasets relative to their census population size is greater than those of Indigenous Americans or any other biogeographical group. Importantly, there was no reported representation of Oceanians in PharmGKB 51. For all groups that have low numbers worldwide, there are likely historical reasons for their relative population sizes being smaller than others, for example, because of attempted genocide or colonisation. As such, it is critical not to exclude these groups from genetics and genomics research, especially based on a justification that there are so few of them across the globe.”

We would like to thank the time and overall positive input from the referees and the editor. With their suggestions and comments, we strongly believe that this paper’s quality has significantly improved. We trust that what we have put together will be reviewed favourably for publication in Cell Genomics.

Referees’ reports, second round of review

Reviewer #1: Thank you for addressing many prior concerns.

The tone of the paper has an overall tone of the field isn’t making progress and perhaps even doing worse. This is not the case. Do we have a long way to go? Absolutely.

There must be more explicit recognition of efforts underway to increase diversity in biomedical research. All of Us, the Mexico City Prospective Cohort Study, Qatar Biobank, and the Human Pangenome project to name a few. These are major efforts that take years (over a decade in many cases) to go from an idea to funding to recruitment, data generation, then analysis and publication. The way forwards is not to say research is failing by continuing with extant data while new efforts are still the data generation phase. The progress must be recognized while continuing the call for increased action by additional people.

Minor comment: Table 2 is missing the much larger European ancestry group.

Reviewer #2: The authors have improved this manuscript a lot. It is now much clearer that the unique contribution of this manuscript lies in integrating PRS, GWAS, and pharmacogenomics, as well as updating the progress made in recent years. However, I still

have serious concerns regarding the technical quality of this manuscript:

I. Lack of Cohesiveness: The manuscript lacks cohesion, with major sections assembled without harmonization. For instance, the figures use varying population, ancestry, or country labels. Here are some examples (though there are many more unmentioned):

- a. Figure 1: Uses both "descent" and "ancestry" without clear reasons.
- b. Figures 2 & 3 use "Asia" as a label, while Figure 4A divides it into "East Asia" and "Central/South Asian."
- c. Population/ancestry/country Label Count: Figure 2 has 6 labels; Figure 4A has 9; Figure 5 has 4; Figure 7 has 5. No reason was given.
- d. Color Inconsistency: Label colors are inconsistent across figures, even for "European/Europe", which appears in most figures.

I understand these inconsistencies result from using different tools and databases, but they give the paper a disjointed appearance.

II. Lack of Supporting Evidence:

- a. Lines 111: "UK Biobank data include >20k individuals with group labels other than 'White/British' who are regularly excluded from genetic analyses but could be useful for contributing GWAS results from more diverse genetic ancestral backgrounds."

Where did the authors get the 20K figure? Most literatures that I'm aware of exclude many more individuals, which actually better support the authors' point. For example, the Neale Lab GWAS in 2017 removed almost 150K individuals, and the Pan UKBB GWAS flagged ~80K as "non-European."

Neale Lab GWAS: <http://www.nealelab.is/blog/2017/9/11/details-and-considerations-of-the-uk-biobank-gwas>

Pan UKBB GWAS: <https://pan.ukbb.broadinstitute.org/>

- b. Lines 127-128: "Recent years have seen an increased proportion of GWAS reporting 'missing' ancestry information. This trend should be recognized as a sign of increasing precision and transparency in human genetics research. It reflects a growing understanding that race and ethnicity are distinct from genetic ancestry, which is crucial for accurate data interpretation and representation."

I understand this is from reviewer 1's suggestion which is sensible. However, it would be helpful for the authors to cite evidence supporting these statements.

III. Direct-to-Consumer (DTC) Research Report:

The authors did not address my primary concern about citing the DTC research report. While the public portion cited is accessible, most of the report is not practically available to the scientific community. This makes it difficult to assess the credibility and limitations of the data.

- a. How was it ensured that countries in each continent were reasonably surveyed for the DTC companies? There could be differences in transparency in and access to company registration information across countries, which could confound the findings. How was this mitigated?
- b. Figure 7: This figure is titled "Global distribution of direct-to-consumer genetic testing companies," but the closest data in the open proportion of report is "Direct-to-Consumer

Genetic Testing Market Share" (3rd figure in the open report). The numbers in the donut chart are similar (61.13%, 17.54%, 14.33%, 5.00%, 2.00% in the report vs. 61%, 17%, 15%, 5%, and 2% in the paper). Are these the same data? If so, why was the title changed from "Market share" to "companies"? If "companies" is correct, the authors need to define what they mean: the number of companies, customers, revenue, or profit?

c. Report Quality Concern: The figure in the open DTC report labels Asia Pacific with 61.13% and North America with 17.54%, but the text in the same report states North America as 61.13%. This clear negligence raises concerns about the report's quality.

d. Reference 64: "Direct-To-Consumer Genetic Testing Market Size, Report 2032" - I cannot find 2032 anywhere on the cited website. Was this a typo?

In my original peer review report, I mistakenly submitted my comments to the authors as comments to the Editor, which were therefore kept from the authors. I'm glad that some of my concerns in the original review report have been resolved. After consulting with the editor, I'm attaching my original report here for the authors' reference. I would appreciate the authors' responses to comments #2a (Figure 1 should now be Figure 2), #3 (Japan), #4, #6 (Page 13 should be page 18 now [line 667]), and #7-9. My comments above (I-III) have also clarified the following comments #5 and #10. I truly apologize for my oversight and the trouble this has caused the authors and the editors.

In this Perspective, the authors discussed current failures and hurdles in achieving a diverse representation of ancestries in current GWAS, clinical trials, pharmacogenomics, and DTC genetic testing. While I fully agree with the authors on the importance of this message, I have concerns about the novelty, contributions, and technical quality of this work. Specifically,

1. The authors are pioneers in advocating for the diversity of genetics. However, it appears that much of the message in this work had been delivered in the authors' earlier work and the work cited by the authors. There doesn't appear to be much fundamentally new information in this Perspective.

2. The analyses of the diversity in clinical trials, pharmacogenomics, and DTC are interesting, yet unfortunately quite crude, relying on external tools without careful examinations.

a. Figure 1 left panel shows that there are precisely 0 participants in GWAS in the entire GWAS catalog from Australia (and most of South America and Africa continents). This is an extraordinary claim and should be examined more carefully. Just taking the figure from GWAS Diversity Monitor as-is appears insufficient.

b. Page 11: Using GPT-4 to estimate the number of individuals of the Indigenous American population is not scientifically acceptable as this number can't be reproduced nor referenced.

3. The drug efficacy and safety conclusions were reached using US-based data from the FDA, thus naturally biased towards the US population composition rather than that globally. For this analysis to be fair and reflect the actual global representation, the authors should perhaps include data from the regulatory bodies of other countries or regions, such as the European Medicines Agency and Pharmaceuticals and Medical Devices Agency (Japan).

4. The authors appear to have restricted the discussions on genetic ancestry. This is fine but perhaps should be made clear and discussed as a limitation because additional factors, such as environment and culture, are also important factors that we need to measure from diverse ancestral populations to fully understand human complex disorders.

5. The ancestral groupings are inconsistent across different sections of this Perspective (e.g., the biogeographic grouping in Figure 3A differs greatly from the ancestry grouping in Figure 4). This is likely due to using existing tools or databases with inconsistent ancestry labels. Nevertheless, this lowers the quality of this paper.

6. The authors claimed that "current strategies for inclusion are insufficient" on Page 13. While this is very likely correct, analyses in the Perspective really showed this bias as a result of tens of years of practice, not of the "current strategies." It's unclear whether recent efforts, some inspired by the authors' earlier work, effectively address the inclusion issues. For example, as the All-of-Us project was developed with inclusion and diversity in mind, does it work better than the earlier biobank efforts such as the UK Biobank? If it is still insufficient, what could have been done to make it better?

7. While the authors acknowledged the limitations in the DTC testing, I feel the concern regarding the utility of DTC versus its risk should be thoroughly presented as due diligence (e.g., <https://www.science.org/content/article/genetics-group-slams-company-using-its-data-screen-embryos-genomes>).

8. Related to #7, some countries have heavily regulated DTC, which might contribute to its low availability to some populations. This reflects an informed choice balancing the risk versus benefit of DTC and should be distinguished from the issues discussed in this Perspective.

9. The increase in the GWAS samples in Europeans can be greatly driven by biobanks. This raises two issues:

- a. To some extent, are we double counting the European samples as they are the same biobank samples driving hundreds, if not thousands, of GWASs?
- b. The utility of biobanks for many diseases is limited, as population- and community-based biobanks are biased towards healthy individuals. The authors perhaps could separate GWASs driven by biobanks versus disease-focused consortia as they can have different issues and be subject to different challenges and strategies.

10. Ref 47 has an access fee of \$4900 and is not typically covered by institutional subscriptions. While for-fee article access is unfortunately typical, a scientific paper usually costs much less with an even cheaper annual subscription. In this case, I wonder if the authors had the opportunity to access and read ref 47 full text to ensure the accuracy of data and appropriateness of the method in producing the cited figure (the figure is freely accessible)? I'm particularly interested in the DTC companies they included outside of the US

and Europe: what was done to ensure this list is as complete as possible.

Reviewer #3: Comments enter in this field will be shared with the author; your identity will remain anonymous.

I am glad to see a quantitative, multi-aspect, approach to documentary the lack of diversity in human genetic studies. The choices of CYP2D6 and of warfarin as examples are particularly well-chosen.

My comments are limited to minor suggestions for improvement.

* The title is a little misleading. I don't think the paper really address "why we're still failing in diversity"

* Although it's not my place to criticize the GWAS Diversity Monitor (shown in Figure 1), it's a little ironic that the projection used by the monitor overemphasizes the area of N America, Europe, and Russia.

* The last sentence ("However, the last ...") in the middle paragraph of page 3 is confusingly written ("greater proportions of missing ancestries . . . with as few as")

* Figure 3A could do with some more explanation. For example, this is the first use of the "American" label, which readers might need some help in recognizing as indigenous American. The super-imposed pie charts are hardly self-explanatory either.

* Page 8, first paragraph, there appears to be a gap (between 1 and 1.25) between intermediate metabolizer and normal metabolizer

* Page 11, last paragraph. The first sentence is a bit hard to understand, and if I am guessing correctly as to what it means, it is contradicted by the next sentence.

11. Minor issue: the two figures on page 2 do not have citations, legends, nor labels. I think they were taken from the authors' previous publications, which is fine, but they should be clearly labeled and explained in the manuscript.

Reviewer #3: I found the authors' changes in response to my previous concerns (and those of the other reviewers) to be more than satisfactory. I have no further concerns and I think the current iteration is quite strong.

Authors' response to the second round of review

Please find attached a new revised version of our manuscript of ref. CELL-GENOMICS-D- 24-00109.R2 entitled "Bridging Genomics' Greatest Challenge: The Diversity Gap", which we submitted to be considered for publication in your journal. We have modified the text and title following the comments and suggestions made by Reviewer 1 and 2. We understand that Reviewer 3 is satisfied with our responses. All new changes and pertinent bits of the responses below have been highlighted in red in the revised manuscript.

Reviewer #1:

Thank you for addressing many prior concerns. The tone of the paper has an overall tone of

the field isn't making progress and perhaps even doing worse. This is not the case. Do we have a long way to go? Absolutely. There must be more explicit recognition of efforts underway to increase diversity in biomedical research. All of Us, the Mexico City Prospective Cohort Study, Qatar Biobank, and the Human Pangenome project to name a few. These are major efforts that take years (over a decade in many cases) to go from an idea to funding to recruitment, data generation, then analysis and publication. The way forwards is not to say research is failing by continuing with extant data while new efforts are still the data generation phase. The progress must be recognized while continuing the call for increased action by additional people.

Thank you for highlighting this important point. We appreciate your recognition for your previous revisions and the time you have taken to clarify your concerns.

We agree that there are ongoing efforts to increase diversity in biomedical research and that these initiatives deserve explicit recognition in our manuscript. We acknowledge that the tone of our original submission might have overly focused and emphasised the gaps and challenges in the field without giving due credit to the progress already been made. To address these concerns, we have done the following.

1. We have added a dedicated section entitled "ONGOING EFFORTS TO INCREASE DIVERSITY" (lines 126-142) to recognise key global initiatives such as All of Us, the Mexico City Prospective Cohort Study, the Qatar Biobank, the Peruvian Genome Project, and the Human Pangenome Project. Here, we describe the scope and current status of such projects. We also make sure that this acknowledgement situates our argument in the proper context of ongoing progress. Moreover, we frame our argument that, although current datasets still lack sufficient diversity, there are several initiatives that represent meaningful progress and must be built upon to accelerate diversity in genomics.

ONGOING EFFORTS TO INCREASE DIVERSITY

Across the globe, significant efforts are being undertaken to enhance diversity in genomics. The All of Us project¹⁶, for instance, has actively recruited participants from various ancestries across the United States to build one of the most diverse health databases in the world. The Mexico City Prospective Cohort Study¹⁷ and the Peruvian Genome Project¹⁸ are defining Latin American initiatives aiming to provide insights into the unique health challenges faced by admixed and native indigenous communities of the Americas. In the Middle East, the Qatar Biobank Cohort Study¹⁹ has broadened the scope of representation for this region. Similarly, the Human Pangenome Project²⁰ is working on sequencing genomes from historically underrepresented populations. These initiatives have focused on collecting diverse genetic data from underrepresented populations to ensure more inclusive and representative genome database references, which will inform better our current landscape of genomic human variation. The data they are generating is steadily contributing to a more inclusive genomics landscape, though much remains to be done to accelerate the progress and ensure broader global representation.

2. We have changed the title to "Bridging Genomics' Greatest Challenge: The Diversity Gap". This is in recognition of your comment that there has been progress to increase diversity in biomedical research. In addition, we expect that the more positive spin in the title provides a more suitable tone for readers.

We believe these changes address your concerns properly and create a more balanced and constructive narrative about the state of diversity in genomics.

1. Minor comment: Table 2 is missing the much larger European ancestry group.

Thanks for bringing this detail to our attention. Unfortunately, statistics about the European Ancestry group have not been provided by the source [<https://research.23andme.com/research-innovation-collaborations/>], hence we are unable to report them accurately. Nevertheless, we have included a note in the legend acknowledging this point (line 448; “[n]umber of European participants is not provided by the source.”).

Reviewer #2:

The authors have improved this manuscript a lot. It is now much clearer that the unique contribution of this manuscript lies in integrating PRS, GWAS, and pharmacogenomics, as well as updating the progress made in recent years.

Thank you for acknowledging our efforts to improve the original submission based on your initial comments. We will continue to address your insightful feedback, for which we are immensely grateful.

However, I still have serious concerns regarding the technical quality of this manuscript:

I. Lack of Cohesiveness: The manuscript lacks cohesion, with major sections assembled without harmonization. For instance, the figures use varying population, ancestry, or country labels. Here are some examples (though there are many more unmentioned):

a. Figure 1: Uses both "descent" and "ancestry" without clear reasons.

a. The reason for this discrepancy as you are well aware is because the study sourced from Popejoy and Fullerton (2016) reused the data that was published by Need and Goldstein (2009), which categorised them as “European descent”. We generated a new figure that homogenises both figures, using the same labels and colours. We strive for consistency in terms of labels, colours and classifications and now provide our own figure design with the same colours (navy blue for “Europeans” and green for “Other”) and entitle the figures as “GWAS Catalog Ancestry Composition”. We hope that these changes adequately address your concerns.

Figure 1. Sampling Bias: Left, Number of genome-wide association study (GWAS) participants of European ancestry in 2009 from the GWAS Catalog. Right, update by Popejoy and Fullerton (2016) 9 on the ancestry breakdown in GWAS studies in 2016. Figure adapted and taken from

9.

b. Figures 2 & 3 use "Asia" as a label, while Figure 4A divides it into "East Asia" and "Central/South Asian."

c. Population/ancestry/country Label Count: Figure 2 has 6 labels; Figure 4A has 9; Figure 5 has 4; Figure 7 has 5. No reason was given.

d. Color Inconsistency: Label colors are inconsistent across figures, even for "European/Europe", which appears in most figures. I understand these inconsistencies result from using different tools and databases, but they give the paper a disjointed appearance.

b. c. d. Thank you for your thoughtful and detailed feedback. We agree that the manuscript may give the impression that it lacks full cohesion, especially regarding population, ancestry and country labels across different figures and sections. As you rightly suggest, this inconsistency arises from the use of various tools, databases and publicly available resources classifying regions, populations and ancestries differently.

Some datasets use broader categories like "Asia", others break this into more specific regions such as "East Asia" and "South Asia". Similarly, "descent" and "ancestry" are sometimes used interchangeably depending on the source. No doubt this lack of standardisation has contributed to the variation in terminology, hence contributing to results appearing disjointed.

To address these issues, we have done the following:

1. We have revised the manuscript to standardise the use of terms like "descent" and "ancestry" across in Figure 1. Where resources use different groupings, we have either harmonised labels or provided clear explanations indicating differences.

2. We have included a dedicated section in DISCUSSION ("Limited Available Data and Inconsistencies in Population Labels", see below; lines 483-526) that discusses the inherent limitations of the available data. These limitations include inconsistencies in population labels across different data resources. Here we also discuss that even though we are striving for cohesion, the variations in how populations are defined by different studies and databases present an important challenge for building a unified narrative.

Limited Available Data and Inconsistencies in Population Labels

A key challenge in our analysis is the inherent variability in how different datasets define and categorise populations. This variability arises from the use of multiple resources and tools, each with their own population labels, ancestry classifications and country groupings. This poses significant barriers to achieving complete cohesion in our analysis and presentation of results.

The genomics databases we analysed, including GWAS, PGx, DTC genetic testing and US FDA drug trials, define populations based on different criteria. For instance, the GWAS Diversity Monitor groups individuals broadly using categories such as 'European', 'Asian', 'African', 'African American or Afro-Caribbean', 'Hispanic or Latin American' and 'Other/Mixed'. Other resources such as PharmGKB, further divide populations into more specific subgroups such as 'East Asian', 'Central/South Asian', 'Near Eastern' or 'Sub-Saharan African'. Similarly, the terms 'ancestry', 'descent' and 'ethnicity' are used interchangeably in some studies but defined more narrowly in others, adding confusion and variability.

Figures 2 and 3 of this manuscript, derived from the GWAS Diversity Monitor, refer to broader geographic categories such as 'Asia'. Figure 4A shows labels as 'East Asian', 'Central/South Asian' and 'Near Eastern', reflective of the different classification system used by PharmGKB. The US FDA drug trial snapshot reports differently populations of African origin, including under the same label 'African and African American'. These differences are not arbitrary and reflect the underlying methodologies of the original datasets. As we strive to present a unified analysis, it is not always possible to align these labels across the manuscript without oversimplifying or misrepresenting the source data.

These challenges also extend to visual representations. We note that 'European' is represented as pink by the GWAS Diversity Monitor (Figures 2 and 3), while PharmGKB represent European as green (Figure 4A). We recognise that this creates a disjointed appearance where the preservation of original colour schemes and groupings are necessary to maintain the integrity of the sources.

It is important to note that some data sources limit their representation to specific populations, leading to underrepresentation of certain regions or ancestries. For instance, the term Oceanian only appears in PharmGKB and it is absent from the GWAS Diversity Monitor, DTC genetic testing and Clinical Trials. This shortcoming is severe for the incumbent population, as it might skew the analysis towards regions or populations where genomic data are more readily available. It is therefore important to acknowledge ascertainment bias in some data sources we rely on, which significantly complicates the task of fully harmonising a global view of genetic diversity. Although these challenges do not undermine the validity of our analysis, they highlight the need for greater diversity awareness and standardisation of mainstream health genomic datasets.

3. Wherever possible, we have adjusted the colour schemes across all figures. We are still in the situation that both the GWAS Diversity Monitor and PharmGKB's biogeographical regions are differently coloured. We have clearly stated it in the legends of Figure 2 and 4 ("We note inconsistencies in labelling and colouring of populations due to different ways of reporting ancestries by sources.").

We hope these changes significantly improve cohesiveness and provide clearer explanations of the unavoidable limitations posed by current available data. Thank you once again for your valuable input.

II. Lack of Supporting Evidence:

a. Lines 111: "UK Biobank data include >20k individuals with group labels other than 'White/British' who are regularly excluded from genetic analyses but could be useful for contributing GWAS results from more diverse genetic ancestral backgrounds." Where did the authors get the 20K figure? Most literatures that I'm aware of exclude many more individuals, which actually better support the authors' point. For example, the Neale Lab GWAS in 2017 removed almost 150K individuals, and the Pan UKBB GWAS flagged ~80K as "non-European." Neale Lab GWAS: <http://www.nealelab.is/blog/2017/9/11/details-and-considerations-of-the-uk-biobank-gwas> Pan UKBB GWAS: <https://pan.ukbb.broadinstitute.org/>

Thank you for your insightful comments regarding the UK Biobank exclusion figures. You are correct that the literature often cites significantly higher exclusion numbers, particularly in large GWAS studies like those from the Neale Lab and Pan UKBB. The 20K figure referenced in our manuscript was intended to reflect a rough estimate of non-White/British participants within the UK Biobank dataset who are frequently excluded from analyses. However, we acknowledge that this number is an underestimation when compared to other reports.

In response to your comment, we have instead referenced the Pan UKBB GWAS number and written (line 113):

'It is also important to note that UK Biobank data contain ~80K "non-European" [...]'.

b. Lines 127-128: "Recent years have seen an increased proportion of GWAS reporting 'missing' ancestry information. This trend should be recognized as a sign of increasing precision and transparency in human genetics research. It reflects a growing understanding that race and ethnicity are distinct from genetic ancestry, which is crucial for accurate data interpretation and representation." I understand this is from reviewer 1's suggestion which is sensible. However, it would be helpful for the authors to cite evidence supporting these statements.

Thank you for the suggestion. We have now included the following 4 references within the paragraph (lines 146-150):

21. Ju, D., Hui, D., Hammond, D. A., Wonkam, A. & Tishkoff, S. A. Importance of Including Non-European Populations in Large Human Genetic Studies to Enhance Precision Medicine. *Annu Rev Biomed Data Sci* 5, 321 (2022).

22. Abdellaoui, A., Yengo, L., Verweij, K. J. H. & Visscher, P. M. 15 years of GWAS discovery: Realizing the promise. *Am J Hum Genet* 110, 179–194 (2023).

23. Rebbeck, T. R., Mahal, B., Maxwell, K. N., Garraway, I. P. & Yamoah, K. The distinct impacts of race and genetic ancestry on health. *Nature Medicine* 2022 28:5 28, 890–893 (2022).

24. Jorde, L. B. & Bamshad, M. J. Genetic Ancestry Testing: What Is It and Why Is It Important? *JAMA* 323, 1089–1090 (2020).

III. Direct-to-Consumer (DTC) Research Report:

The authors did not address my primary concern about citing the DTC research report. While the public portion cited is accessible, most of the report is not practically available to the scientific community. This makes it difficult to assess the credibility and limitations of the

data.

a. How was it ensured that countries in each continent were reasonably surveyed for the DTC companies? There could be differences in transparency in and access to company registration information across countries, which could confound the findings. How was this mitigated?

b. Figure 7: This figure is titled "Global distribution of direct-to-consumer genetic testing companies," but the closest data in the open proportion of report is "Direct-to- Consumer Genetic Testing Market Share" (3rd figure in the open report). The numbers in the donut chart are similar (61.13%, 17.54%, 14.33%, 5.00%, 2.00% in the report vs. 61%, 17%, 15%, 5%, and 2% in the paper). Are these the same data? If so, why was the title changed from "Market share" to "companies"? If "companies" is correct, the authors need to define what they mean: the number of companies, customers, revenue, or profit?

c. Report Quality Concern: The figure in the open DTC report labels Asia Pacific with 61.13% and North America with 17.54%, but the text in the same report states North America as 61.13%. This clear negligence raises concerns about the report's quality.

d. Reference 64: "Direct-To-Consumer Genetic Testing Market Size, Report 2032" – I cannot find 2032 anywhere on the cited website. Was this a typo?

Thank you for your thorough review of the Direct-to-Consumer (DTC) research report section. Based on your detailed concerns, we have reconsidered our use of this report and its associated figure.

We acknowledge that the accessibility and reliability of the cited report pose significant challenges to assessing the credibility and limitations of the data. Given these issues and the inconsistencies you highlighted in the report's contents, we have eliminated the section and Figure 7 altogether. We believe this is the best course of action to maintain the integrity of our manuscript. We sincerely thank you for bringing these concerns to our attention.

In my original peer review report, I mistakenly submitted my comments to the authors as comments to the Editor, which were therefore kept from the authors. I'm glad that some of my concerns in the original review report have been resolved. After consulting with the editor, I'm attaching my original report here for the authors' reference. I would appreciate the authors' responses to comments #2a (Figure 1 should now be Figure 2), #3 (Japan), #4, #6 (Page 13 should be page 18 now [line 667]), and #7-9. My comments above (I-III) have also clarified the following comments #5 and #10. I truly apologize for my oversight and the trouble this has caused the authors and the editors.

In this Perspective, the authors discussed current failures and hurdles in achieving a diverse representation of ancestries in current GWAS, clinical trials, pharmacogenomics, and DTC genetic testing. While I fully agree with the authors on the importance of this message, I have concerns about the novelty, contributions, and technical quality of this work. Specifically,

1. The authors are pioneers in advocating for the diversity of genetics. However, it appears that much of the message in this work had been delivered in the authors' earlier work and the work cited by the authors. There doesn't appear to be much fundamentally new information in this Perspective.

2. The analyses of the diversity in clinical trials, pharmacogenomics, and DTC are interesting, yet unfortunately quite crude, relying on external tools without careful examinations.

a. Figure 1 left panel shows that there are precisely 0 participants in GWAS in the entire GWAS catalog from Australia (and most of South America and Africa continents). This is an extraordinary claim and should be examined more carefully. Just taking the figure from GWAS Diversity Monitor as-is appears insufficient.

Thank you for your valuable feedback regarding the map in the original figure. We agree that the "Participants by country (all parent terms)" map could be misinterpreted due to its focus on specific years in which data were produced. Since the map reflects participants based on the timing of the studies rather than their actual genetic diversity or ancestry, it can create confusion about the geographic representation of participants. For example, some countries may appear underrepresented or overrepresented in certain years due to when studies were conducted, rather than providing a more accurate and holistic view of global participation across all studies.

Given this potential for misinterpretation, we have decided to remove the map altogether from Figure 2. In its place, we have included a new figure:

Figure 2. GWAS Diversity Monitor: Left, Number of genome-wide association study (GWAS) participants by ancestry, including different types of GWAS studies or health conditions (parent terms), discovery stages, 2023. Right, Number of GWAS participants across all parent terms, discovery stage, 2024. [Accessed online 9 Sept 2024]. We note inconsistencies in labelling and colouring of populations between figures due to different ways of reporting ancestries by sources.

This new figure focuses on the participants' ancestry and the overall inclusion across all parent terms within the GWAS Diversity Monitor. We believe this approach provides a clearer and transparent view of the diversity in GWAS participants.

b. Page 11: Using GPT-4 to estimate the number of individuals of the Indigenous American population is not scientifically acceptable as this number can't be reproduced nor referenced.

Agreed. We removed GPT4 estimates (addressed in the previous revision).

3. The drug efficacy and safety conclusions were reached using US-based data from the FDA, thus naturally biased towards the US population composition rather than that globally. For this analysis to be fair and reflect the actual global representation, the authors should perhaps include data from the regulatory bodies of other countries or regions, such as the European Medicines Agency and Pharmaceuticals and Medical Devices Agency (Japan).

To address global bias in our analysis, we agree it would be helpful to integrate any available demographic data from the European Medicine Agency (EMA) and other international agencies to complement the FDA data. This would help ensure conclusions reflect more global insights into drug efficacy and safety. We have found that direct ancestry breakdowns are not readily available from EMA trials. We were not able to find a specific, consolidated breakdown of ancestry representation in EMA-regulated clinical trials. EMA guidelines do suggest that ethnic diversity is being increasingly prioritised, but how effectively this is implemented remains variable across studies (please see <https://www.coreclinicalsciences.com/newsletter/ema-trial-diversity> and <https://toolbox.eupati.eu/resources/inclusion-diversity-clinical-trials/>).

In response to your request regarding the ancestry breakdown in the Pharmaceuticals and Medical Devices Agency (PMDA) studies in Japan, specific demographic details, particularly those related to ancestry, are not as easily accessible as in other regions like the US. Most clinical trials overseen by the PMDA are focused on maintaining quality standards and meeting regulatory requirements for the Japanese population, and historically, trials have primarily involved participants of Japanese ancestry (see <https://ascpt.onlinelibrary.wiley.com/doi/full/10.1111/cts.12485>). The inclusion of broader ethnic groups in PMDA trials, especially within Japan, is not as emphasized compared to regions like the US and Europe but efforts are increasingly been made (see https://www.jstage.jst.go.jp/article/ghm/4/4/4_2022.01007/_article).

We consider that paragraphs between lines 298 and 314 properly address these limitations (highlighted in red in the main text for your convenience):

In addition to utilising FDA data, we also examined resources from the European Medicines Agency (EMA) ⁴⁴, ClinicalTrials.gov ⁴⁵, and the World Health Organization (WHO) ⁴⁶. The EMA provides data on clinical trials conducted within Europe, ClinicalTrials.gov aggregates information from clinical trials conducted worldwide, and the WHO International Clinical Trials Registry Platform (ICTRP) ⁴⁷ compiles data from various international registries. However, none of these resources offer summary statistics on participant demographics, including ancestry. EMA, ClinicalTrials.gov and WHO require reviewing each study individually to determine if demographic data are available, and even then, there is no assurance that such data will be included.

The absence of readily accessible demographic information for these studies poses a significant barrier to addressing and reducing health disparities across different ancestries. Researchers who seek to conduct demographic data analyses to track and monitor disparities in diversity and inclusion of the resources must extract and compile the data manually, which hinders efforts toward equitable representation in clinical trials. The generalisability of research findings thus continues to be limited across diverse populations, and it is often unclear to whom they are (and are not) applicable.

4. The authors appear to have restricted the discussions on genetic ancestry. This is fine but

perhaps should be made clear and discussed as a limitation because additional factors, such as environment and culture, are also important factors that we need to measure from diverse ancestral populations to fully understand human complex disorders.

Thank you for this valuable comment. The primary reason we have restricted our discussion to genetic ancestry is because the manuscript focuses specifically on the role of genetic ancestry in the context of genomics. Our intention is to provide a targeted analysis of how underrepresentation in genomic studies impacts on the interpretation and application of genetic data. While we fully acknowledge that other factors such as environment, culture, and socioeconomic conditions are crucial for a comprehensive understanding of human health and disease, the scope of this paper was deliberately narrowed to address the diversity gaps in genetic ancestry representation.

That said, we agree that these additional factors are part of a much wider context and are essential for fully understanding complex disorders. Following the referee's advice, we have clarified in the new manuscript version that our focus on genetic ancestry should be aligned in future work integrating environmental and cultural variables for a more holistic approach.

We have added this text at the end of DISCUSSION to address your point (lines 688-691):

However, we recognise that environmental, cultural, and socioeconomic factors are integral to fully understanding human diversity. Future research should seek to integrate these broader contextual elements for a more holistic approach to understanding diversity within precision medicine and healthcare equity.

5. The ancestral groupings are inconsistent across different sections of this Perspective (e.g., the biogeographic grouping in Figure 3A differs greatly from the ancestry grouping in Figure 4). This is likely due to using existing tools or databases with inconsistent ancestry labels. Nevertheless, this lowers the quality of this paper.

Thank you for your observation regarding the inconsistency in ancestral groupings across different sections of the paper. You are correct that the variation stems from the use of different tools and databases, each of which categorises ancestry labels differently. This is an important challenge in the field, as there is no universal standard for grouping genetic ancestry, and each database often uses its own conventions based on geographic or biogeographic criteria.

We acknowledge that this inconsistency can impact on the clarity and cohesion of the paper. To address this, we have included a dedicated section ("Limited Available Data and Inconsistencies in Population Labels"; lines 483-526), as stated above, discussing the limitations arising from the use of different resources, and how the lack of harmonisation in ancestry labels across datasets can affect interpretations.

6. The authors claimed that "current strategies for inclusion are insufficient" on Page 13. While this is very likely correct, analyses in the Perspective really showed this bias as a result of tens of years of practice, not of the "current strategies." It's unclear whether recent efforts, some inspired by the authors' earlier work, effectively address the inclusion issues. For example, as the All-of-Us project was developed with inclusion and diversity in mind, does it work better than the earlier biobank efforts such as the UK Biobank? If it is still insufficient, what could have been done to make it better?

Thank you for your valuable feedback. We agree that biases in genomic diversity are the result of historical practices over many years, and addressing these long-standing gaps remains a challenge. However, the data presented in Figure 2, which provides the most up-to-date information on GWAS diversity, along with the additional analyses we have performed, suggest that current strategies for inclusion are still insufficient. While recent initiatives such as the All-of-Us project and other efforts are aimed at increasing diversity, the overall proportional representation of non-European ancestries remains significantly low.

In our analyses, even though newer projects are prioritising inclusion, the scale and speed at which diverse populations are being recruited, do not seem to be keeping pace with the vast numbers of European-centric datasets, such as those derived from the UK Biobank. While All-of-Us is making strides compared to earlier efforts, our findings show that the representation from historically underrepresented populations is still minimal in many large genomic studies, indicating that more needs to be done to overcome these challenges.

7. While the authors acknowledged the limitations in the DTC testing, I feel the concern regarding the utility of DTC versus its risk should be thoroughly presented as due diligence (e.g., <https://www.science.org/content/article/genetics-group-slamscompany-using-its-data-screen-embryos-genomes>).

Thank you for your comment. We understand the concerns surrounding the utility and ethical risks of direct-to-consumer (DTC) genetic testing. While DTC testing is mentioned in our analysis, our intent is not to delve deeply into the ethics or risks associated with its use, as that would require a more focused exploration in an ethics-centred discussion.

This paper aims to highlight the lack of diversity in genomic datasets, including those from DTC testing, rather than assess the broader implications of DTC technologies. Although the risks of DTC testing are important and worth exploring, we have chosen to restrict our discussion to the specific issue of quantifying genetic ancestry, as this aligns with the primary focus of our manuscript.

8. Related to #7, some countries have heavily regulated DTC, which might contribute to its low availability to some populations. This reflects an informed choice balancing the risk versus benefit of DTC and should be distinguished from the issues discussed in this Perspective.

Thank you for your insightful comment. We acknowledge that the regulation of direct-to-consumer (DTC) genetic testing varies significantly between countries, which can indeed impact its availability to certain populations. In countries with strict regulations, such as Germany, France, and Italy, these laws are often designed to protect consumers from potential risks like privacy concerns, misinterpretation of genetic information, or lack of medical guidance. As a result, the low availability of DTC services in these regions is often an informed regulatory decision to balance the potential risks and benefits.

We agree that this regulatory constraint should be distinguished from the broader issue discussed in this perspective, which focuses primarily on the lack of diversity in genomic data across different populations. While regulations influence the accessibility of DTC testing in certain regions, the core issue of underrepresentation in genetic research extends beyond DTC testing to include large-scale genomic studies and biobank-driven projects. We have

clarified this distinction in the new manuscript version highlighting that although regulatory factors do play a role in DTC testing availability, the primary concern remains the insufficient inclusion of diverse populations in broader genomic datasets.

We have added this paragraph in DISCUSSION (lines 632-640) and included two new references to support it:

In several countries, such as Germany, France, and Italy, strict regulations on direct-to-consumer (DTC) genetic testing limit its availability due to concerns about privacy, misinterpretation, and the absence of medical guidance^{80,81}. These regulations are designed to protect consumers but reduce access compared to regions with more lenient laws like the U.S. However, it is important to distinguish this issue from the broader lack of diversity in genomic research, which remains a significant challenge across large-scale studies. The focus of this paper is on the underrepresentation of non-European populations in global genomic datasets, which persists independently of DTC testing regulations.

80. Kalokairinou, L. et al. Legislation of direct-to-consumer genetic testing in Europe: a fragmented regulatory landscape. J Community Genet 9, 117–132 (2018).

81. Borry, P. et al. Legislation on direct-to-consumer genetic testing in seven European countries. European Journal of Human Genetics 2012 20:7 20, 715–721 (2012).

9. The increase in the GWAS samples in Europeans can be greatly driven by biobanks. This raises two issues:

a. To some extent, are we double counting the European samples as they are the same biobank samples driving hundreds, if not thousands, of GWASs?

b. The utility of biobanks for many diseases is limited, as population- and communitybased biobanks are biased towards healthy individuals. The authors perhaps could separate GWASs driven by biobanks versus disease-focused consortia as they can have different issues and be subject to different challenges and strategies.

Thank you for your important comments. We agree that the increase in GWAS samples of European ancestry is largely driven by biobanks, particularly large-scale resources like the UK Biobank.

a. Double Counting of European Samples:

You raise a valid concern about the potential for double counting European samples due to their repeated use across multiple studies. Biobanks such as the UK Biobank and others provide data that is leveraged for a wide range of GWAS, meaning that the same set of samples may be used in hundreds, if not thousands, of studies. This can inflate the representation of European individuals in the global GWAS landscape, further skewing diversity metrics. We addressed this point in the manuscript in the previous revision, which we have indicated in red font for your convenience (lines 196-202). Here acknowledge that the repeated use of biobank data can lead to an overestimation of European ancestry representation in genomic research (particularly the UK population) and may obscure the scale of underrepresentation of non-European ancestries.

Despite its usefulness, the GWAS Diversity Monitor may report a participant count that can exceed the actual population, due to its methodology. Since each individual is counted in every study they are part of, in 2021 the GWAS Diversity Monitor showed 3675.9 million participants in the United Kingdom, which has a population of 67.0 million. This reflects repeated counts of the same individuals across different traits and phenotypes. Such double counting may make diversity worse than it is, as the absolute number of diverse genomes is increasing²¹.

b. Utility of Biobanks vs. Disease-Focused Consortia:

We acknowledge that population- and community-based biobanks are often biased towards healthier individuals, which limits their utility in studying certain diseases or health conditions that may be more prevalent in underrepresented populations. Disease-focused consortia, on the other hand, are specifically designed to target individuals with specific conditions, thus offering a more relevant dataset for understanding diseasespecific genetic variation. In the revised manuscript, we clarify the distinction between GWAS driven by biobanks and those conducted by disease-focused consortia. We have added a new section in DISCUSSION describing the different challenges each faces, such as population bias in biobanks and the smaller sample sizes often seen in diseasefocused efforts. This distinction provides a clearer understanding of the varying contributions of each to the overall picture of genomic diversity.

Increase of GWAS Samples of Europeans Driven by Biobanks

A key distinction in genomic studies arises from different approaches taken by biobank-driven GWAS and those conducted by disease-focused consortia. A major limitation of biobank-driven GWAS is the potential overrepresentation of certain ancestral groups such as Europeans. This can skew findings towards this population¹⁴. Such overrepresentation can lead to double counting of the same samples across hundreds or even thousands of GWAS due to these datasets being used repeatedly across many studies.

Disease-focused consortia gather data from individuals affected by specific conditions, often including severe disease not well represented in biobanks. These studies tend to involve smaller sample sizes due to the rarity of the diseases being studied, offering more targeted insights into the conditions. Diseases-focused consortia may include more diverse populations, especially if they are related to conditions more prevalent in underrepresented groups^{12,34}. Their smaller sample sizes and narrower focus, however, can limit their generalisability. To address these issues, future research would require both population-based and disease-focused consortia. The integration of both approaches will improve global representation in genome research.

10. Ref 47 has an access fee of \$4900 and is not typically covered by institutional subscriptions. While for-fee article access is unfortunately typical, a scientific paper usually costs much less with an even cheaper annual subscription. In this case, I wonder if the authors had the opportunity to access and read ref. 47 full text to ensure the accuracy of data and appropriateness of the method in producing the cited figure (the figure is freely accessible)? I'm particularly interested in the DTC companies they included outside of the US and Europe: what was done to ensure this list is as complete as possible.

Thank you for your comment. As indicated above, we have removed the section that referenced this source altogether. Upon further review, we determined that the data and the

report in question were not sufficiently reliable. As a result, we have eliminated this reference and its associated analysis from the manuscript to ensure accuracy and integrity in our reporting.

11. Minor issue: the two figures on page 2 do not have citations, legends, nor labels. I think they were taken from the authors' previous publications, which is fine, but they should be clearly labeled and explained in the manuscript.

Thank you for your feedback. We have addressed this issue by updating the Figure 1, as indicated in our earlier response. This figure now uses consistent labels and colours (navy blue for “Europeans” and green for “Other”) to homogenise data sourced from Popejoy and Fullerton (2016) and Need and Goldstein (2009).

Final Comment to Reviewer #2:

We would like to extend our sincere thanks for your thorough and meticulous review of our manuscript. The feedback you provided has helped us refine key areas of the manuscript, particularly addressing inconsistencies, improving the cohesiveness of our figures, and clarifying complex issues such as the impact of biobanks on GWAS data. We truly appreciate the time and effort you have dedicated to reviewing our work, and we believe that the manuscript has been significantly strengthened thanks to your review.

Reviewer #3:

I found the authors' changes in response to my previous concerns (and those of the other reviewers) to be more than satisfactory. I have no further concerns and I think the current iteration is quite strong.

Thank you very much for your positive feedback and for taking the time to review our manuscript. We are pleased to hear that our revisions have addressed your previous concerns satisfactorily and that you find the current iteration strong.

Final Response:

We would like to sincerely thank all the reviewers for their thorough and constructive feedback throughout the revision process. Their insightful comments and suggestions have been invaluable in helping us refine and strengthen our manuscript. We appreciate the time and effort each reviewer has dedicated to improving the quality and clarity of this work. We strongly believe that the revisions made in response to their thoughtful feedback have significantly enhanced the manuscript, and we hope it will be now suitable for publication in Cell Genomics.

We look forward to hearing from you and thank you once again for your careful consideration.

Referees' reports, third round of review

Reviewer #1: Thank you for addressing my concerns in this revision.

Reviewer #2: The authors have made substantial edits to the manuscript. I would like to thank the authors for their efforts. The manuscript has improved a lot and I have no further concerns.

Authors' response to the third round of review

None